# Ice nucleation ability of ammonium sulfate aerosol particles internally mixed with secondary organics

Barbara Bertozzi[1], Robert Wagner[1], Junwei Song[1], Kristina Höhler[1], Joschka Pfeifer[2], Harald Saathoff[1], Thomas Leisner[1], and Ottmar Möhler[1]

[1]Institute of Meteorology and Climate Research, Karlsruhe Institute of Technology, Karlsruhe, Germany
[2]CERN, Geneva, 1211, Switzerland

**Correspondence:** Robert Wagner (robert.wagner2@kit.edu)

**Abstract.** The abundance of aerosol particles and their ability to catalyze ice nucleation are key parameters to correctly understand and describe the aerosol indirect effect on the climate. Cirrus clouds strongly influence the Earth's radiative budget, but their effect is highly sensitive to their formation mechanism, which is still poorly understood. Sulfate and organics are among the most abundant aerosol components in the troposphere and have also been found in cirrus ice crystal residuals.

Most of the studies on ice nucleation at cirrus cloud conditions looked at either purely inorganic or purely organic particles. However, particles in the atmosphere are mostly found as internal mixtures, the ice nucleation ability of which is not yet fully characterized.

In this study, we investigated the ice nucleation ability of internally mixed particles composed of crystalline ammonium sulfate (AS) and secondary organic material (SOM) at temperatures between $-50\,°C$ and $-65\,°C$. The SOM was generated from

the ozonolysis of $\alpha$-pinene. The experiments were conducted in a large cloud chamber, which also allowed to simulate various aging processes that the particles may experience during their transport in the atmosphere, like cloud cycling and redistribution of the organic matter. We found that the ice nucleation ability of the mixed AS/SOM particles is strongly dependent on the particle morphology. Small organic mass fractions of $5-8\,\mathrm{wt\%}$ condensed on the surface of AS crystals are sufficient to completely suppress the ice nucleation ability of the inorganic component, suggesting that the organic coating is evenly distributed

on the surface of the seed particles. In this case, the ice nucleation onset increased from a saturation ratio with respect to ice $S_{ice} \sim 1.30$ for the pure AS crystals to $\geq 1.45$ for the SOM-coated AS crystals. However, if such SOM-coated AS crystals are subjected to the mentioned aging processes, they show an improved ice nucleation ability with the ice nucleation onset at $S_{ice} \sim 1.35$. We suggest that the aging processes change the particle morphology. The organic matter might redistribute on the surface to form a partially engulfed structure, where the ice nucleating active sites of the AS crystals are no longer completely

masked by the organic coating, or the morphology of the organic coating layer might transform from a compact to a porous structure.

Our results underline the complexity to represent the ice nucleation ability of internally mixed particles in cloud models. They also demonstrate the need to further investigate the impact of atmospheric aging and cloud processing on the morphology and related ice nucleation ability of internally mixed particles.

## 1 Introduction

Cirrus clouds form in the upper troposphere and are composed of ice particles. Frequent occurrence, long persistence and large spatial extent make their radiative effect significant for the total Earth's energy budget (Fleming and Cox, 1974; Matus and L'Ecuyer, 2017). At temperatures lower than $-38\,°C$, ice crystals can either form by freezing of diluted aqueous solution particles (homogeneous ice nucleation) or by heterogeneous ice nucleation on a solid particle surface that lowers the nucleation barrier. Heterogeneous ice nucleation can occur via different modes depending on temperature, relative humidity, and particle characteristics (*e.g.*, size, morphology, and chemical composition). The heterogeneous nucleation can occur, for example, via deposition nucleation on the surface of a solid particle, via pore condensation and freezing, and via immersion freezing (Vali et al., 2015). The microphysical and optical properties of cirrus clouds depend on the predominance of the ice formation process which varies with atmospheric dynamics, thermodynamic conditions, as well as type and abundance of aerosol particles (Patnaude and Diao, 2020). Detailed knowledge of the ice formation processes is therefore crucial to correctly formulate and model the microphysical properties of cirrus clouds and their effect on the present and future climate. This knowledge is enhanced by an increasing number of aerosol-cloud interaction studies with particles of different origins and chemical compositions, both in the field and the laboratory (Hoose and Möhler, 2012; Kanji et al., 2017).

Knowing which particles are present in the upper troposphere allows us to identify the candidates involved in cirrus cloud formation. Airborne measurements performed with a single particle mass spectrometer showed organics and sulfate to be the most abundant aerosol components in the upper troposphere (Murphy et al., 2006; Froyd et al., 2009). Most of the particles were found to be internal mixtures, and the sulfate component was often partially or fully neutralized. The particle's organic mass fraction ranged from $30\,\mathrm{wt\%}$ to $70\,\mathrm{wt\%}$, depending on the altitude and the air mass origin (Froyd et al., 2009). A recent study, spanning 10 years of filter measurements, showed a dominant contribution of sulfate or organics depending on location, altitude, and season (Martinsson et al., 2019). In a recent study, Höpfner et al. (2019) detected and identified crystalline ammonium nitrate particles in the upper troposphere during the Asian monsoon period.

In this article, we focus on the ammonium sulfate-organics system and therefore briefly summarize here the current knowledge on the ice nucleation ability of the individual compounds and their internal mixtures. The ice nucleation ability of ammonium sulfate at cirrus conditions depends on the particles' phase state. Liquid ammonium sulfate particles nucleate ice homogeneously at saturation ratios which are in good agreement with predictions of the water-activity-based ice nucleation formulation by Koop et al. (2000), *e.g.* ice onset at the saturation ratio with respect to ice $S_{\mathrm{ice}} = 1.55$ at $-51\,°C$ (Bertram et al., 2000). Crystalline ammonium sulfate particles, however, which form below the efflorescence relative humidity of $40 - 45\,\%$ (Gao et al., 2006), can induce heterogeneous ice formation below about $-50\,°C$, *e.g.* at $S_{\mathrm{ice}} = 1.20$ at $-52\,°C$ (Abbatt et al., 2006). At these low temperatures, the deliquescence relative humidity of ammonium sulfate is higher than the onset relative humidity for heterogeneous ice nucleation, which then occurs at much lower ice saturation ratios than homogeneous freezing of fully deliquesced aqueous particles (e.g., Abbatt et al., 2006; Shilling et al., 2006; Wise et al., 2010). This behavior is also observed for other inorganic salts like sodium chloride (Wise et al., 2012; Wagner and Möhler, 2013) and ammonium nitrate (Wagner et al., 2020).

The phase state is also a critical parameter for the ice nucleation ability of organic particles. Some organic compounds, e.g. oxalic acid, are able to crystallize and act as ice nucleating particles in the immersion freezing (Zobrist et al., 2006; Wagner et al., 2015) and deposition nucleation mode (Kanji et al., 2008). Recently, atmospheric secondary organic aerosol particles have been found in a glassy phase state at low temperatures and/or relative humidity (Virtanen et al., 2010; Shiraiwa et al., 2017) and glassy particles have shown to promote heterogeneous ice formation (Murray et al., 2010; Wilson et al., 2012). These observations have led to an increased number of studies focusing on the ice nucleation ability of secondary organic particles originating from different gas precursors (*e.g.*, Wang et al., 2012; Ladino et al., 2014; Ignatius et al., 2016; Charnawskas et al., 2017; Wagner et al., 2017; Wolf et al., 2020). Heterogeneous ice nucleation on glassy particles was mainly observed in experiments with single organic compounds, whereas the majority of the studies performed with secondary organic aerosol particles from the oxidation of $\alpha$-pinene showed ice formation only at or above the homogeneous freezing threshold (Möhler et al., 2008). Organic material derived from $\alpha$-pinene, also used in our study, is a common proxy for atmospheric organic particles in laboratory experiments (e.g., Möhler et al., 2008; Kanji et al., 2019).

Only a few studies explored the ice nucleation ability of internally mixed particles composed of organic compounds and crystalline ammonium sulfate (AS). Shilling et al. (2006) showed that, at temperatures between $-33\,^{\circ}\mathrm{C}$ and $-83\,^{\circ}\mathrm{C}$, the heterogeneous ice nucleation onset of pure crystalline AS was not altered by the addition of maleic acid in a $1:1$ ratio by weight. Wise et al. (2010), Schill and Tolbert (2013), and Baustian et al. (2013) investigated how the addition of palmitic acid, organic polyols, sucrose, citric acid, and glucose changed the heterogeneous ice nucleation ability of crystalline AS in an environmental cell coupled with an optical microscope and a Raman spectrometer. The organic to inorganic particle mass ratio in these studies varied from $1:1$ to $2:1$. These studies show that the organic components only minimally affect the efflorescence and deliquescence relative humidity of the AS particles (at temperatures between $-33\,^{\circ}\mathrm{C}$ and $-3\,^{\circ}\mathrm{C}$), and also that their intrinsic heterogeneous ice nucleation ability almost remains unchanged (at temperatures between $-63\,^{\circ}\mathrm{C}$ and $-43\,^{\circ}\mathrm{C}$). More recently, Ladino et al. (2014) and Schill et al. (2014) investigated the effect on the ice nucleation ability when more atmospherically relevant organic components are mixed with AS, namely secondary organic material (SOM) from the oxidation of $\alpha$-pinene and from the reaction of methylglyoxal with methylamine. Ladino et al. (2014) probed the ice nucleation ability of AS particles mixed with $\alpha$-pinene SOM with inorganic to organic mass ratios of $1:1$ and $4:1$ at $-55\,^{\circ}\mathrm{C}$. The mixed particles were generated by atomizing a liquid solution of AS mixed with the water soluble fraction of the SOM. The particles passed a diffusion dryer and were size selected ($170\,\mathrm{nm}$) for the ice nucleation measurement. The ice saturation ratio required for $0.1\,\%$ of the mixed particles to nucleate ice was found to be $1.40$, a value between the nucleation onsets of the pure AS particles (at $S_{\mathrm{ice}} = 1.25$) and the pure organic particles ($S_{\mathrm{ice}} = 1.52$). Schill et al. (2014) started their ice nucleation experiments with liquid solution droplets composed of methylglyoxal, methylamine, and AS, which were deposited on a silica disk. Particles were then exposed to low relative humidity to induce efflorescence of the AS component. Optical microscope images revealed that the AS crystallized in numerous isolated islands embedded in the organic matrix. For the mixed particles, the ice onset at $-58\,^{\circ}\mathrm{C}$ occurred at $S_{\mathrm{ice}} = 1.36$, which is below the ice onset saturation ratio for pure organic particles ($S_{\mathrm{ice}} = 1.52$). The authors suggested that immersion freezing on the crystalline ammonium sulfate islands was the main ice nucleation mode for this system.

In this study, we investigated the ice nucleation ability of internally mixed particles composed of crystalline ammonium sulfate and $\alpha$-pinene secondary organic material. The experiments were conducted in the AIDA (Aerosol Interaction and Dynamics in the Atmosphere) cloud chamber, a unique platform to investigate phase changes of aerosol particles during temperature and relative humidity cycles or during cloud processing (Wagner et al., 2017). Formation and dissipation of clouds, as well as cycles of humidification and drying are examples of processes commonly occurring in the atmosphere, but their effects on the physical properties of the so-processed aerosol particles are still poorly understood. In particular, we studied the effect of several aging and cloud processing mechanisms on the particles' ice nucleation ability. The ice nucleation ability was either tested in situ by expansion cooling experiments in the AIDA chamber, or ex situ by sampling the particles and probing them in a continuous flow diffusion chamber. At first, we conducted two types of reference experiments, shown in the schematic of Fig. 1 as experiments of type A and B. In experiment A, we tested the ice nucleation ability of pure crystalline ammonium sulfate (AS) particles. In experiment B, we used an external aerosol preparation chamber to generate crystalline AS particles thickly coated with secondary organic material (SOM) derived from $\alpha$-pinene. The organic coating completely suppressed the heterogeneous ice nucleation ability of the AS core. Based on these data, we then performed dedicated coating experiments in the AIDA chamber, where the organic coating thickness was gradually increased to quantify the effect of the organic material condensing on the AS surface (experiments of type C).

Particles investigated in experiments of type B and C probably had a core-shell morphology, with the ice nucleation (IN) active ammonium sulfate core shielded by the condensed, not IN active organic material. However, phase state and morphology of the particles present in the atmosphere might differ from such idealized core-shell geometries. Mixtures of organics and ammonium sulfate can also form at higher temperature and relative humidity conditions at which the particles are fully mixed, as e.g. in liquid clouds. In such a situation, the important question is: what would be the particle morphology if the crystallization of the ammonium sulfate component is only induced at a later stage, when the particle is subjected to a sufficiently low relative humidity? Would this process also inevitably lead to evenly coated particles where the heterogeneous IN activity of the AS component is strongly suppressed? An important process that needs to be considered is that initially homogeneously mixed aqueous particles can undergo the so-called liquid-liquid phase separation when the relative humidity is reduced (Marcolli and Krieger, 2006). For mixed particles of AS and organics, several studies have determined the relative humidity at which the liquid-liquid phase separation occurs (separation relative humidity) as function of the temperature, the organic-to-sulfate mass ratio, and the oxygen-to-carbon ratio (O : C) (e.g., Ciobanu et al., 2009; Bertram et al., 2011). For example, Bertram et al. (2011) showed that for mixed AS/SOM derived from $\alpha$-pinene, isoprene, and $\beta$-caryophyllene at 290 K and with O : C $< 0.5$, the separation relative humidity is at about $90\%$. The phase separated particles can assume different morphologies depending on temperature, drying rate, viscosity of the organic component (Fard et al., 2017), and size of the particles (You et al., 2014; Freedman, 2020). Apart from a core-shell arrangement, possible particle morphologies also include several islands of the inorganic component in the organic matrix or partially engulfed structures. If the relative humidity is then further reduced to induce the crystallization of the AS, the ice nucleation active inorganic component is not necessarily completely surrounded by the deactivating SOM, with strong implications for the particles' ice nucleation ability. We investigated this processing pathway in experiment D. We started with the thickly SOM-coated AS particles from the reference experiment B, but then activated them

to cloud droplets in the AIDA chamber at a higher temperature ($-5\ ^\circ$C), thereby generating homogeneously mixed aqueous droplets. We then reduced the relative humidity to induce the AS crystallization and we probed the particles' ice nucleation ability at cirrus cloud conditions.

Another pathway that can lead to a morphology change in particles of organic-inorganic mixtures is cloud processing in a convective cloud system. Adler et al. (2013) showed that aerosol particles composed uniquely of natural organic matter, or mixed with ammonium sulfate, assume a highly porous morphology after an atmospheric freeze-drying process that involved droplet activation, freezing, and sublimation of ice crystals. The effect was more visible for the purely organic compared to the internally mixed particles. Wagner et al. (2017) simulated such freeze-drying process by an expansion cooling experiment in the AIDA chamber with secondary organic aerosol particles derived from the ozonolysis of $\alpha$-pinene and also observed the formation of highly porous particles. In experiment type E, we repeated the freeze-drying experiment from Wagner et al. (2017) with the thickly SOM-coated AS particles to investigate whether the expected change in morphology would also affect the particles' ice nucleation ability. The effect might be twofold: first, the porous organic coating could, by itself, be more ice nucleation active than the non-porous coating due to the recently proposed pore condensation and freezing mechanism (Marcolli, 2014); second, a porous coating layer could less efficiently shield the ice nucleating active AS component from the environment, thereby also enhancing the particles' ice nucleation ability after the freeze-drying process.

## 2  Experimental methods

### 2.1  General setup

AIDA (Aerosol Interaction and Dynamics in the Atmosphere) is a cloud simulation chamber which has extensively been described elsewhere (e.g., Möhler et al., 2003), so only a brief description is provided here. Figure 2 presents a schematic of the experimental setup used for this study. The AIDA chamber mainly consists of an aluminium vessel ($84.3\ \mathrm{m}^3$ volume) located inside a thermostatic housing. Air ventilation through heat exchangers inside the housing allows to control the temperature of the cloud chamber from $+60\ ^\circ$C to $-90\ ^\circ$C (accuracy $\pm 0.3\ ^\circ$C). Mechanical pumps allow to control and reduce the chamber pressure down to $0.01$ hPa, with different pumping speeds. The chamber is cleaned by flushing several times with clean and dry synthetic air at low pressures between 1 and 10 hPa. Before re-filling to ambient pressure with synthetic air, ultra-pure water is evaporated into the evacuated chamber to add the amount of water required for the following experiment. With this procedure, the walls can be partially covered with an ice layer to maintain ice-saturated conditions and to provide sufficient water vapor during the expansion cooling experiments (described in Sect. 2.3.1). The water vapor concentration inside the chamber is measured with an in situ multi-path tunable diode laser absorption spectrometer (APicT) with an accuracy of $\pm 5\ \%$ (Fahey et al., 2014). The water saturation ratios with respect to water and ice are computed by using the liquid water and ice saturation pressures from Murphy and Koop (2005).

A second vessel is located in proximity to the AIDA chamber. It is called aerosol preparation and characterization (APC) chamber, is made of stainless steel, has a volume of $3.7\ \mathrm{m}^3$, is operated at ambient temperature, can also be evacuated down to 1 hPa and flushed with synthetic air for cleaning purposes, and is re-filled with clean and dry synthetic air before starting the

experiments. The APC chamber allows to prepare the aerosol particles in a reproducible way (i.e., under well-controlled and repeated experimental conditions) and to transfer them into the AIDA chamber. Furthermore, the evacuated APC chamber can be used as a buffer volume for cloud simulation experiments at a high expansion rate in the AIDA chamber (more details are provided in Sect. 3.3).

Standard aerosol particle instrumentation is present at the facility to measure the aerosol number concentration and number size distribution. Both the AIDA and the APC chambers are equipped with condensation particle counters (models 3010 and 3022 from TSI) and scanning mobility particle sizers (SMPS, TSI). Two optical particle counters (OPC, model welas, Palas GmbH) are installed at the bottom of the AIDA chamber to measure the size distribution of large aerosol particles, cloud droplets, and ice crystals in two different size ranges ($0.7 - 46$ µm and $5 - 240$ µm, Wagner and Möhler (2013)). The two OPCs are located inside the thermal housing to prevent evaporation of cloud droplets and sublimation of ice crystals. The additional instrumentation used to characterize the aerosol population in terms of chemical composition and phase state is described in the next section.

## 2.2   Aerosol preparation and characterization

The aerosol particles were generated, characterized, and aged in the AIDA chamber and in the aerosol preparation and characterization (APC) chamber. The coating procedure performed in the APC chamber aimed at generating a thick organic coating layer on the AS seed particles with high reproducibility. The ice nucleation ability of these particles was either directly probed in a reference experiment (experiments of type B) or after subjecting the particles to different processing steps (experiments of type D and E). The coating experiments performed in the AIDA chamber, instead, aimed at quantifying the effect of a thin coating on the inorganic seeds (experiments of type C). In this case, the coating procedure consisted of several steps, each with a small amount of organics condensing on the seed aerosol particles. The coating procedure in the APC chamber was performed at ambient temperature and low relative humidity, and in the AIDA chamber at low temperature and ice saturated conditions. Table 1 provides a list of the performed experiments together with information on the thermodynamic conditions at which the particles were generated, the concentration of the gas precursors, the median diameter of the seed particle size distributions before the coating procedure, and the organic mass fraction after the coating procedure.

An aqueous solution was prepared by dissolving ammonium sulfate (Merck, $99.5$ %) in ultra-pure water to obtain a $1$ wt% solute concentration. The solution was then aerosolized by means of an ultrasonic nebulizer (GA 2400, SinapTec). A series of diffusion dryers (Topas GmbH) ensured a relative humidity with respect to water in the injection line lower than $1$ %, well below the efflorescence relative humidity of ammonium sulfate (Onasch et al., 1999; Gao et al., 2006). Secondary organic material (SOM) was generated from the ozonolysis of $\alpha$-pinene (Aldrich, $99$ %). Another important pathway for the formation of $\alpha$-pinene SOM in the atmosphere are photo-oxidation reactions, but the ice nucleating ability of the organic material generated via dark ozonolysis or photooxidation of $\alpha$-pinene was found to be very similar (Piedehierro et al., 2021). Ozone was produced with a silent discharge generator (Semozon 030.2, Sorbios) in pure oxygen.

The generation of AS particles with thick SOM coating was started by injecting crystalline AS particles ($\sim 10^5$ particles cm$^{-3}$) into the APC chamber. Then ozone was added ($\sim 2$ ppm), followed by a continuous flow of $\alpha$-pinene for approximately 30

minutes (experiments B1 and D2) or 1 hour (experiments B2, D1 and E1), resulting in a total concentration between 100 and 300 ppb of injected precursor gas (see Table 1 for the concentrations used in each experiment). The high initial aerosol concentration in the APC chamber during the $\alpha$-pinene injection provided a sufficiently large total surface area for the organic reaction products to condense on the existing AS particles, and to suppress, in the majority of cases, the nucleation and growth of pure organic aerosol particles. At the end of the coating procedure, a fraction of the aerosol particles were transferred into the AIDA chamber.

For the thin coating experiments performed in the AIDA chamber (experiments of type C), the coating was performed in three to four steps, with the injection of smaller amounts of $\alpha$-pinene. Each step consisted in the injection of 100 to 700 ppt of organic gas precursor, summing up to a total concentration of approximately 1 ppb (see Table 1 for the concentrations used in each experiment). A new injection step of gas precursor was performed approximately every hour, to allow the reaction to be completed and to simultaneously perform the ice nucleation measurements (see Sect. 3.1).

A high resolution time of flight aerosol mass spectrometer (HR-ToF-AMS, Aerodyne) measured the chemical composition of the particles, with an uncertainty of 5.3 % regarding the measured mass concentrations (based on ionization efficiency calibration). The particles' organic content was measured after the transfer of the particles from the APC chamber to the AIDA chamber (experiments type B, D, and E) or continuously during the thin coating experiments in AIDA (experiments of type C). During the coating procedure of experiments B2 and C2, some pure organic particles nucleated and eventually grew to a size detectable in the SMPS and the mass spectrometer. For these experiments, to obtain the organic mass concentration of the organic coating on the AS particles, the total organic mass measured by the AMS was corrected for the mass of the smaller mode of nucleated pure organic particles. This was done by independently deriving the total mass concentration of the smaller particle mode from the size distribution measured with the SMPS (estimating the density of the organic material as $\rho_{\text{org}} = 1.25 \text{ g cm}^{-3}$ (Saathoff et al., 2009)) and then subtracting it from the total mass concentration measured with the AMS. The organic mass fraction values $f_{\text{org}}$ reported in Table 1 refer to the organic content at the end of the coating procedures, i.e., regarding experiments of type C, they refer to the organic content after the last coating step. The size-resolved measurements of the chemical composition of the particles by the AMS indicate that the organic mass fraction is constant and not a function of the particle size.

To estimate the thickness of the organic coating, $d$, we combined the size distribution of the pure crystalline AS particles from the SMPS measurements and the organic mass fraction $f_{\text{org}}$ from the AMS. Thereby, we assume the crystalline AS seeds to be spherical particles with diameter $D_p$, and the organic material to be evenly distributed on their surface, leading to a spherical organic shell. The coating thickness $d$ is thus calculated assuming a perfectly concentric core shell morphology as follows:

$$d = \frac{D_p^{\text{coated}} - D_p}{2} \tag{1}$$

The diameter of the coated particles $D_p^{\text{coated}}$ is calculated by considering the size-dependence of the particles' organic mass $M_{\text{org}}(D_p)$:

$$D_p^{\text{coated}}(D_p) = \left[ D_p^3 + \frac{6}{\pi} \frac{M_{\text{org}}(D_p)}{\rho_{\text{org}}} \right]^{\frac{1}{3}}, \tag{2}$$

with:

$$M_{\text{org}}(D_p) = \frac{f_{\text{org}}}{1 - f_{\text{org}}} \cdot \rho_{\text{AS}} \cdot \frac{\pi}{6} D_p^3 \tag{3}$$

The resulting estimated organic coating thickness corresponding to the median diameter of the particle population is reported in Table 1. In Table 1, we also indicate between brackets the estimated coating thickness for seed particles with diameters of 300 and 500 nm, i.e., the size range which comprises the major particle mode in the number size distribution (see below and Fig. 3 for the number size distribution measurements). The thickness estimated for the thickly SOM-coated AS particles (experiments of type B, D and E) ranged from 24 nm to 54 nm. Thinner coatings were obtained at the end of the coating experiments performed in the AIDA chamber (experiments of type C) with 6 nm, 5 nm, and 8 nm coating thicknesses.

Exemplary particle size distributions for the pure crystalline AS and for the SOM-coated AS particles are reported in Fig. 3. The effect of the organic coating on the particle diameter is clearly visible for the thickly coated particles (Fig. 3a, experiment D1). Instead, no change is detectable in the SMPS measurements for the thin coating experiments, in which a lower amount of organic material was produced and condensed on the particles (Fig. 3b, experiment C3). However, due to the numerous processes taking place during the coating procedures (e.g., change of the particles' shape factor) we cannot directly infer the coating thickness from the SMPS measurements.

The phase state of the ammonium sulfate content in the mixed particles (i.e., crystalline or liquid) has been determined with depolarization measurements of back-scattered laser light and the analysis of infrared extinction spectra recorded by means of a Fourier transform infrared spectrometer (FTIR, type IFS66v, Bruker) (Wagner et al., 2006). The presence or absence of specific liquid water absorption bands and changes in the peak position and band width of the $\nu_3$ sulfate mode at about $1100 \, \text{cm}^{-1}$ between crystalline and liquid AS allowed to infer the phase state of the AS content in the mixture (Onasch et al., 1999; Zawadowicz et al., 2015). The SIMONE instrument (Schnaiter et al., 2012) measures the intensity of the light scattered by the aerosol particles in the forward ($2\,°$) and backward ($178\,°$) directions from a linearly polarized laser ($\lambda = 488 \, \text{nm}$). In the backward direction, the polarization-resolved scatter light intensity is detected so that the linear depolarization ratio of the aerosol particles can be determined. Non-spherical and inhomogeneous particle morphologies, like a crystalline core of AS or islands of crystalline AS in the mixed particles, can induce a change of the incident polarization state of the laser, thereby causing a non-zero value for the depolarization ratio. In contrast, homogeneously mixed, spherical aqueous solution droplets would show a depolarization ratio of zero. A detailed example of the use and interplay of the SIMONE and FTIR data in the interpretation of our experiments is given in Sect. 3.2.

### 2.3 Ice nucleation measurements

The ice nucleation ability of the aerosol particles was probed by means of expansion cooling experiments inside the AIDA chamber and with a continuous flow diffusion chamber sampling the aerosol from the AIDA chamber. Details about these two methods are provided below.

#### 2.3.1 Expansion cooling experiments

Two vacuum pumps evacuate the AIDA chamber with a controlled and variable evacuation rate. The related pressure reduction induces an expansion of the gas inside the chamber and thus a temperature drop. The drop in temperature, associated with an almost linear decrease of the water partial pressure due to pressure reduction, results in an increase in the ice saturation ratio that eventually leads to ice nucleation. The pressure drop is typically $200 - 250 \, \mathrm{hPa}$, corresponding to a temperature drop of $5 - 10 \, ^\circ\mathrm{C}$. The AIDA ice nucleation experiments presented in this study were performed at cirrus cloud conditions, with starting temperatures between $-45 \, ^\circ\mathrm{C}$ and $-60 \, ^\circ\mathrm{C}$. Ice starts to nucleate as soon as the critical ice supersaturation for heterogeneous and/or homogeneous ice formation is reached, which depends on the temperature and the aerosol particle type. The nucleated ice quickly grows to large crystals and by that also deplete the water vapor present in the chamber. The optical particle counters detect a fraction of the aerosol particle population as well as the ice crystals. The ice crystal number concentration is evaluated by defining a size threshold to distinguish between the smaller inactivated aerosol particles and the larger nucleated ice crystals. The ice nucleating fraction (INF) is the fraction of aerosol particles that nucleates ice. It is defined as the ratio of the ice crystal number concentration to the total aerosol particle number concentration measured with the CPC. However, when purely organic particles nucleated during the coating procedure, the particle number concentration is evaluated from integration of the SMPS data excluding the nucleation mode of pure organic particles. In fact, as already shown by previous AIDA measurements (Wagner et al., 2017), these purely organic particles do not contribute to the INF of a heterogeneous ice nucleation mode, but induce ice formation only at or above the homogeneous freezing threshold.

#### 2.3.2 Continuous Flow Diffusion Chamber

The Ice Nucleation Instrument of the Karlsruhe Institute of technology (INKA, Schiebel (2017)) is an adapted version of the cylindrical continuous flow diffusion chamber (CFDC) designed by Rogers (1988). Two concentric copper cylinders ($150 \, \mathrm{cm}$ high) form a narrow gap ($\sim 1 \, \mathrm{cm}$) where a total flow of $10 \, \mathrm{L \, min^{-1}}$ passes through. The aerosol sample flow represents $5 \, \%$ of the total flow, while the remaining part is particle free dry synthetic air, that ideally confines the sampled particles to a thin cylindrical laminar flow. This confinement ensures that the particles experience almost the same thermodynamic conditions. The range of conditions that the aerosol lamina experiences represent a source of uncertainty. The instrument walls are kept at different temperatures to induce a diffusional flow profile of heat between the walls. Furthermore, the walls are covered with a thin ice layer to generate a diffusional flow profile of water vapor, driven by the different water saturation vapor pressure above the two ice layers kept at different temperatures. Thus, because of the non-linear dependency of the equilibrium vapor pressure as a function of temperature, a supersaturation profile establishes between the walls.

The temperature of the walls is controlled with two external chillers (LAUDA Proline 890, minimum operational bath temperature $-90\,°$C). The mean temperature of the inner and outer walls is calculated from a set of thermocouples located at different heights along the columns. By increasing the temperature difference between the two walls, it is possible to increase the supersaturation experienced by the aerosol lamina, keeping at the same time its temperature constant. A home-built software allows to program humidity scans by controlling the bath temperature of the chillers.

The upper part of the instrument is called nucleation and growth section and it covers two thirds of the length of the instrument. In this upper section, the walls are kept at different temperatures to generate defined supersaturated conditions with respect to ice. The lower third of the instrument is called droplet evaporation section. In this lower part of the instrument, sub-saturated conditions with respect to water are obtained by maintaining the walls at the same temperature (equal to the cold wall temperature). The evaporation section minimizes droplet and ice crystal coexistence when the instrument is operated at mixed-phase cloud conditions. When operated at cirrus conditions, the evaporation section does not influence the measurements because the environment is at ice saturated conditions ($S_{ice} = 1$), preventing the sublimation of the ice crystals. Additionally, homogeneous freezing of solution droplets in the cold evaporation section is not expected to occur in the experiments presented here ($T \leq -45\,°$C). Temperature and relative humidity experienced by the particle flow are calculated using the approach from Rogers (1988). The errors associated with the calculated temperature and relative humidity of the sample flow are evaluated according to Schiebel (2017). Uncertainties in the wall temperature measurements ($\pm 0.5\,°$C) and in the estimate of the gap width between the two columns ($\pm 0.08\,$cm) are included in the calculations to obtain upper and lower estimates of temperature and water saturation at the aerosol lamina position. The ice crystal number concentration is measured at the instrument outlet by means of an optical particle counter (CLIMET CI-3100). Aerosol particles and ice crystals are distinguished assessing a threshold in the optical particle counter data. The total particle number concentration entering the instrument is measured with a condensation particle counter (CPC-3072, TSI). When organic particles nucleated during the coating procedure, the particle number concentration is evaluated from integration of the SMPS data excluding the nucleation mode of the smaller, purely organic particles.

To avoid frost formation in the cold inlet of the instrument, the temperature inside INKA was limited in relation to the frost point temperature of the AIDA chamber. Measurements have been mainly performed between $-46\,°$C and $-60\,°$C. During the experiments of type D, where the aerosol particles were processed at a higher temperature in the AIDA chamber ($-5\,°$C), a diffusion dryer was added ahead of the instrument inlet to prevent its clogging.

## 3 Results and discussion

The results from the different types of experiments are presented in three different subsections. The reference experiments with pure and thickly SOM-coated crystalline AS particles, as well as the gradual thin coating experiments performed in the AIDA chamber are presented first (Sect. 3.1). The results on the ice nucleation ability of initially thickly SOM-coated AS crystals, which were temporarily activated to homogeneously mixed, aqueous droplets and then recrystallized are presented in Sect. 3.2

(experiment D). Finally, the experiments where the initially thickly SOM-coated AS crystals were subjected to freeze-drying processing in the AIDA chamber are presented in Sect. 3.3 (experiment E).

## 3.1 Organic coating effect (experiments A, B, and C)

The ice nucleation ability of pure and SOM-coated solid ammonium sulfate particles at cirrus cloud conditions was probed with experiments of type A, B, and C (Fig. 1). The particles were coated in the APC chamber for experiments of type B and in the AIDA chamber for experiments of type C. In both cases, we assume that the coating procedure led to particles with a core-shell morphology, with the solid AS as the core and the organic material surrounding it. This hypothesis is supported by the ice nucleation measurements as discussed below. We conducted the experiment of type B two times with different coating time and amount of $\alpha$-pinene added to the APC chamber. In experiment B1, we generated SOM-coated AS particles with an organic mass fraction of 26.9 % and a coating thickness of 28 nm; in experiment B2, the organic mass fraction was 39.1 % and the coating thickness 50 nm. As explained in Sect. 2.2, these coating thicknesses were computed for the median particle diameters assuming a uniform coating.

Time series of the AIDA expansion cooling runs for experiments A1 (pure crystalline AS) and B2 are presented in Fig. 4. The upper panels show the pressure and temperature drop during the expansion runs. The middle panels show the saturation ratio with respect to ice ($S_{\text{ice}}$) and the homogeneous freezing threshold (Koop et al., 2000). The lower panels depict the fraction of aerosol particles activated as ice crystals (ice nucleating fraction, INF). Black dotted vertical lines indicate when the INF reaches the threshold of 0.1 %, defined here as the ice nucleation onset. Crystalline AS particles induce heterogeneous ice nucleation at saturation values lower than those required for the homogeneous freezing of solution droplets. For crystalline AS the ice onset is measured at $S_{\text{ice}} = 1.29$ (Fig. 4a middle panel). For SOM-coated AS particles, instead, the ice onset is measured at $S_{\text{ice}} = 1.44$ (Fig. 4b middle panel). The organic coating clearly suppresses the ice nucleation activity of the crystalline AS core and shifts the ice nucleation onset close to the homogeneous freezing threshold. The strong decrease of the particles' ice nucleation ability suggests that most of the IN active sites of the bare AS have been shielded by the condensed organic material distributed in a core-shell morphology.

Figure 5a summarizes the ice onset saturation ratios for pure and thickly SOM-coated crystalline AS particles from this study (experiments A1, B1, and B2). The water saturation ($RH_{\text{w}} = 100$ %), the homogeneous freezing threshold for $\Delta a_{\text{w}} = 0.3$ (Koop et al., 2000), and the AS deliquescence relative humidity (Clegg et al., 1998) are also shown in the figure. Results from AIDA and INKA are shown as squares and circles, respectively, and the symbol colors refer to the different aerosol types used in the different experiments (orange for pure AS particles and light and dark green for experiments B1 and B2, respectively). Both methods clearly show the suppression of the heterogeneous ice nucleation activity of the crystalline ammonium sulfate core by the organic coating. The AIDA results show lower ice onset saturation ratios for both pure and SOM-coated AS particles compared to the INKA results. Wagner et al. (2020) already noted this discrepancy in previous experiments with inorganic particles at cirrus conditions and encouraged inter-comparison studies of different methods and instruments at low temperatures. It is worth mentioning that secondary organic material is more viscous at low temperature and this also affects water diffusion. Therefore, different experimental techniques could be subjected to different kinetic limitations, inducing ice

nucleation at different conditions. During an AIDA expansion run, for example, the thermodynamic conditions inside the chamber change relatively slow and allow long equilibration times. However, inside the continuous flow diffusion chamber, new particles are continuously sampled, suddenly exposed to a supersaturated environment, and only have a few seconds ($\sim 10$ s) to activate as ice crystals.

In Fig. 5b, we compare our data to literature results of $S_{ice}$ at the ice nucleation onset for pure crystalline AS particles (red colored symbols) and pure secondary organic aerosol (SOA) particles (green colored symbols). One can see that the pure AS and pure SOA data form clearly separated blocks on the $S_{ice}$ vs. $T$ diagram, with the heterogeneous nucleation onsets for AS ranging from 1.1 to 1.3 and for pure SOA from 1.4 to 1.7. With regard to our experiments with the thickly SOM-coated AS particles, experiment B1 shows a slightly lower ice onset saturation ratio ($S_{ice} = 1.4$, AIDA experiment) compared to the measurements of the pure SOA particles. However, this early onset was only representative of a small fraction of the aerosol particle population and if we had chosen an INF of 1 % as the onset condition, the related $S_{ice}$ value had already increased to 1.46. When analyzing in detail the results of the processing experiments on the particles' ice nucleation ability (experiments of type D and E), we will not only compare the onset conditions but show the full spectrum of the INF as a function of $S_{ice}$.

Experiments B1 and B2 have shown that thick SOM-coating layers almost completely suppress the heterogeneous ice nucleation ability of the AS component and shift the particles' ice nucleation onset close to that observed for the pure SOA particles. We thus conducted experiments of type C to investigate and quantify the effect of a thin organic coating layer on the ice nucleation activity of crystalline AS particles. As described in Sect. 2.2, the coating procedure for these experiments was performed directly in the AIDA chamber at different temperatures ($-45$ °C, $-50$ °C, and $-60$ °C) and at ice saturated conditions by stepwise oxidizing small amounts of $\alpha$-pinene. After each coating step, the aerosol particles were sampled and their ice nucleation ability measured with the continuous flow diffusion chamber at the same, or slightly lower, temperature of the AIDA chamber ($\sim -50$ °C, $\sim -54$ °C, and $\sim -60$ °C). Figure 6a shows the ice nucleating fraction (INF) as a function of the saturation ratio with respect to ice for humidity ramps performed at $-54$ °C with particles characterized by different organic mass fractions with INKA. The activation curves from experiments C1 and C3 are shown with solid symbols (circles and down-pointing triangles, respectively), data from experiments B1 and B2 are shown as empty circles. The gradual shift of the ice onset towards higher saturation values with the increase of the condensed organic material is clearly visible. Already after the second coating step of experiment C1, that corresponded to an organic mass fraction of only 2.4 wt%, most of the ice crystals that had formed at $S_{ice} < 1.4$ on the pure crystalline AS particles have disappeared from the records. After the fourth coating step of experiment C1, when the organic mass fraction has reached a value of 5.7 wt%, the ice nucleation onset was shifted close to the value observed for pure $\alpha$-pinene SOA particles. The ice nucleation ability of particles with $\sim 8$ wt% organic material (experiment C3) is almost identical to the one of the thickly coated particles from experiments B1 and B2 (with 27 wt% and 39 wt% organic mass fractions, respectively), indicating that such a low organic content, if uniformly distributed, is sufficient to suppress the ice nucleation ability of the solid AS particles. The progressive increase in the ice onset saturation ratios measured by the continuous flow diffusion chamber highlights its sensitivity to detect the effect of the organic coating.

Figure 6b summarizes the ice onset saturation ratios, corresponding to an INF of 0.1 %, obtained from the INKA measurements as function of the organic mass fraction of the particles. Results for experiment A1 (left side) and for all experiments of

type C (right side) are shown. Different symbols correspond to different experiments, colors indicate at which temperature the ice nucleation ability was measured. The data confirm the results from the experiments presented in Fig. 6a, showing that the ice nucleation ability of the solid AS component is gradually suppressed with the increase of organic material condensed on its surface. In most cases organic mass fractions of only $4-8\,\mathrm{wt\%}$ (yielding estimated coating thicknesses of $5-8\,\mathrm{nm}$) shift the ice nucleation onsets of the coated AS particles to values above $S_{\mathrm{ice}} = 1.45$ at temperatures between $-45\,°\mathrm{C}$ and $-55\,°\mathrm{C}$, i.e. to the regime where also pure SOA particles would nucleate ice. These small organic mass fractions were sufficient to completely suppress the heterogeneous ice nucleation ability of the crystalline AS core. The suppression of the particles' ice nucleation ability during and at the end of the coating procedure suggests, as for experiments of type B, that the organic material is evenly distributed on the AS surface and progressively covers its IN-active sites.

The temperature trend of the ice nucleation onsets for organic mass fractions larger than about $4\mathrm{wt\%}$ revealed by experiments C, i.e., higher $S_{\mathrm{ice}}$ values with decreasing temperature, could point to a homogeneous ice nucleation pathway of the coated particles, meaning that at least the outer layer of the organic material has liquefied during the particles' residence time in INKA. A similar temperature trend has been observed in ice nucleation studies with pure $\alpha$-pinene SOA particles (e.g., Ladino et al., 2014; Wagner et al., 2017; Charnawskas et al., 2017). Furthermore, we need to consider that the condensed organic material could have a different chemical composition and viscosity at the different temperatures of the in situ coating experiments (Huang et al., 2018), thereby affecting the water diffusion into the particles and the ice nucleation pathways (Berkemeier et al., 2014; Lienhard et al., 2015; Price et al., 2015; Fowler et al., 2020).

## 3.2 Liquid cloud processing (experiment D)

In view of the results from Sect. 3.1, that the ice nucleation onset saturation ratios of SOM-coated crystalline AS particles are already at or above $S_{\mathrm{ice}} = 1.45$ for small organic mass fractions of $10\,\mathrm{wt\%}$, one might ask the question whether this type of internally mixed particles should still be considered as candidates for inducing heterogeneous ice nucleation in cloud models. Furthermore, mass spectrometer measurements by Froyd et al. (2009) show that the average organic mass fraction of internally mixed particles is even higher (ranging from $30$ to $70\,\mathrm{wt\%}$). Experiments of type C, however, just tested one specific pathway for the formation of the mixed particles, where the crystalline AS was already present and then coated with the organic substances. As already discussed in the introduction, different particle morphologies, like partially engulfed structures, might form when we start from homogeneously mixed, aqueous mixtures of sulfate and organics and then induce the crystallization of the AS component. This pathway was investigated with experiments of type D, where liquid cloud processing is simulated. Here, we started from thickly SOM-coated AS crystals as investigated in experiments of type B, temporarily activated these particles to homogeneously mixed, aqueous droplets in a short expansion run conducted in the AIDA chamber at warm and humid conditions, and then reduced the relative humidity to induce the re-crystallization of the ammonium sulfate component.

Figure 7a shows a schematic of the liquid cloud processing together with the possible phase states and morphologies of the particles during the experiment. The thickly coated ammonium sulfate particles were generated as described in Sect. 2.2 in the APC chamber (stage I in Fig. 7a) and then transferred into the AIDA chamber (stage II), which was held at $-5\,°\mathrm{C}$ and $RH_{\mathrm{w}} = 85\,\%$. These are conditions where the ammonium sulfate component is already in a liquid state (the AS deliquescence

relative humidity at $-5\,°\mathrm{C}$ is $83\,\%$ $RH_\mathrm{w}$ (Clegg et al., 1998)), but where the entire particle is probably still in a liquid-liquid

phase separated state (separation relative humidity can be as high as $\sim 90\,\%$ (You et al., 2014)). Figure 7b presents the time series of the AIDA pressure (black line), AIDA temperature (red line), AIDA relative humidity with respect to water (blue line), and linear depolarization ratio measured with the SIMONE instrument (green line) during the liquid cloud processing experiment. The liquid state of the ammonium sulfate fraction upon transfer from the APC chamber into the AIDA chamber is verified by the FTIR measurements, reported in Fig. 7c. Spectrum A (in orange) is a reference spectrum of crystalline

ammonium sulfate particles from Exp. A1 in the regime of the $\nu_3$ $(\mathrm{SO}_4^{2-})$ mode at about $1114\,\mathrm{cm}^{-1}$. Spectrum B (in blue) was recorded after addition of the organic-coated AS crystals into the AIDA chamber. Here, the $\nu_3$ mode is clearly broadened and shifted to lower wavenumbers. These spectral changes are indicative of the transition of AS from the solid to the liquid state (Zawadowicz et al., 2015). Also the low value of the depolarization ($\delta \sim 1\,\%$) is indicative of a particle morphology that is close to that of a homogeneous sphere. However, the nonzero value of $\delta$ indicates that some particle inhomogeneity still remains. We

suggest that the particles are characterized by two phase-separated, slightly eccentrically arranged liquid components, because $\delta$ would also be zero for concentric spheres (Bohren and Huffman, 1998).

To then transform the phase-separated particles in the AIDA chamber into homogeneously mixed liquid particles, we performed a short expansion run in order to reach $RH_\mathrm{w} = 100\,\%$ and thereby exceed the relative humidity threshold for the liquid-liquid phase separation (stage III in Fig. 7a). During the expansion run, the depolarization $\delta$ indeed dropped to the back-

ground value of about $0\,\%$ when the relative humidity increased above $90\,\%$, indicating that we had successfully transformed the initially SOM-coated AS crystals to homogeneously mixed aqueous droplets. After the expansion, we refilled the AIDA chamber to ambient pressure and induced the recrystallization of the AS component by reducing the relative humidity to $32\,\%$ (stage IV). This was done by extracting the humid air from the chamber and refilling it at the same rate with dry synthetic air. Note that in this specific experiment the chamber walls were not ice-coated, because the ice layer would have acted as a source

of water vapor and prevented the reduction of the relative humidity. To support the reduction in relative humidity, we also slightly warmed the AIDA chamber to $-3.5\,°\mathrm{C}$. The recrystallization of the AS fraction becomes evident from the increase of the depolarization $\delta$ over time and the associated change in the infrared signature of the particles. Spectrum C in Fig. 7c (in purple), which was recorded at $RH_\mathrm{w} = 32\,\%$ at the end of the recrystallization step, clearly shows that the $\nu_3$ mode of particles has shifted back to the peak position typical for crystalline AS. This procedure has been applied in two experiments

with particles of different organic mass fraction (experiments D1 and D2, see Table 1).

The ice nucleation ability of the in situ crystallized particles has been probed with INKA at $-54\,°\mathrm{C}$ immediately after the AS crystallization (for experiments D1 and D2), and with an expansion cooling experiment after overnight cooling of the AIDA chamber to $-50\,°\mathrm{C}$ (for experiment D2). As the INKA measurements were performed by sampling from the AIDA chamber at $-5\,°\mathrm{C}$ and $RH_\mathrm{w} \sim 32\,\%$, a diffusion dryer was used to prevent frost formation in the instrument inlet. To infer

the possible effect of the dryer on the ice nucleation ability of the recrystallized particles, an additional INKA measurement was performed after the cooling of the AIDA chamber, thus without the need of a diffusion dryer. The ice nucleation ability measured with and without the dryer (i.e., before and after the cooling) are comparable (not shown), indicating that the dryer did not influence the ice nucleation ability of the particles. Figures 7d and 7e present the ice nucleating fraction (INF) as a

function of the saturation ratio with respect to ice $S_{ice}$ for pure AS crystals (orange data, experiment A1), unprocessed thickly

SOM-coated AS crystals (green, experiment B1), and for the internally mixed particles that were subjected to the droplet

activation and in situ crystallization process (yellow data, experiment D2). Very similar results were obtained for experiment

D1. The AIDA and INKA results indicate that the ice nucleation ability of the processed particles lies between those of the

pure crystalline AS and of the thickly SOM-coated AS crystals. The onset of the heterogeneous ice nucleation mode of the

processed particles is observed at $S_{ice} \sim 1.35$. The in situ crystallization of initially homogeneously mixed aqueous AS/SOM

particles can therefore lead to particle morpholgies that are heterogeneously ice nucleation active, even if the organic mass

fraction is as high as $39.1\,\mathrm{wt\%}$ or $24.3\,\mathrm{wt\%}$ (Exps. D1 and D2, respectively). Most probably, the particles adopt a partially

engulfed structure, where some ice nucleating sites of the AS component remain uncovered by the organic material (Freedman,

2020). This result again supports the hypothesis of a core-shell morphology for the coated particles of experiments B, for which

the heterogeneous ice nucleation ability of the AS component was completely masked by the condensed organic material.

The measured ice onsets agree with previous measurements of Ladino et al. (2014) and Schill et al. (2014). As noted in the

introduction, Ladino et al. (2014) investigated mixed particles composed of AS and the water soluble fraction of SOM derived

from $\alpha$-pinene. The particles were generated from liquid solutions with inorganic to organic mass ratios of $1:1$ and $4:1$, dried,

and then probed in a CFDC on their ice nucleation ability at $-55\,^{\circ}\mathrm{C}$. They also found an intermediate ice nucleation onset of

the mixed particles (at $S_{ice} = 1.4$) compared to reference experiments with pure crystalline AS (ice onset at $S_{ice} = 1.25$) and

pure secondary organic particles (ice onset at $S_{ice} = 1.52$). The results from Ladino et al. (2014) are shown for comparison

with the INKA data in Fig. 7e (triangles, AS in orange, mixed particles in yellow, and pure SOA in green). The results for the

pure SOA particles from Ladino et al. (2014) and for the thickly SOM-coated particles from this study agree, confirming that

the coated particles have an ice nucleation behavior similar to the purely organic particles, consistent with our assumption of a

core-shell morphology. Also the ice nucleation behavior of the mixed particles obtained from homogeneously mixed aqueous

AS/SOM solutions upon crystallization of the AS component nicely agree. Schill et al. (2014) have shown that complex

particle morphologies are formed when AS crystallizes from solution droplets composed of methylglyoxal, methylamine, and

AS. In their experiment, the ice nucleation onset of the internally mixed particles was lower compared to the pure organics, in

agreement with our results.

We note, however, that the underlying heterogeneous ice nucleation mode might be different in the various studies. In

addition to deposition nucleation, crystalline AS particles might also be ice nucleation active in the immersion freezing mode

(Zuberi et al., 2001). This activity may be controlled by the viscosity and water solubility of the organic material (Schill

and Tolbert, 2013; Schill et al., 2014). The finding that $\alpha$-pinene SOM mass fractions greater than $\sim 5-10\,\mathrm{wt\%}$ completely

suppress the heterogeneous ice nucleation ability of crystalline AS (Sect. 3.1) rules out that immersion freezing is a prevalent

nucleation mode in our experiments. The higher heterogeneous ice nucleation activity of internally mixed AS/SOM particles

formed by cloud droplet activation and recrystallization of the AS component must therefore be related to a change in the

particle morphology to a partially engulfed structure. For organic materials with a lower viscosity and higher water solubility,

however, the mixed particles might also be heterogeneously ice nucleation active in a core-shell morphology due to immersion

freezing by the crystalline AS core. Schill et al. (2014) assumed that immersion freezing was indeed responsible for the

heterogenous ice nucleation mode observed in their experiments. As Ladino et al. (2014) only used the water soluble fraction of the generated SOM, its hygroscopic behavior might be different compared to our study so that their observed heterogeneous ice nucleation mode of the mixed particles after the AS crystallization might also be due to immersion freezing. The viscosity and amount of the organic material will also determine whether the crystallization of AS in internally mixed aqueous AS/SOM particles can occur at all. In particular, at lower temperatures and/or higher amounts of organics, the AS efflorescence could also be inhibited (Bodsworth et al., 2010). If re-crystallization of the AS component and the formation of a partially engulfed particle morphology did not occur at lower temperatures (in contrast to the $-5\,°\mathrm{C}$ condition simulated in this study), liquid cloud processing would not increase the ice nucleation ability of the AS/SOM particles. This behavior could be investigated in future AIDA experiments where the re-crystallization process is performed at lower temperatures.

The enhanced ice nucleation ability of the liquid cloud processed particles compared to the pure organic and organic-coated particles clearly indicates that the ice nucleation ability of atmospheric aerosol particles can strongly change during their lifetime in the atmosphere. Cloud processing of internally mixed aerosol particles is a common phenomenon, whose impacts on the particles' microphysical properties need to be investigated in future studies.

### 3.3 Freeze-drying processing (experiment E)

After an atmospheric freeze-drying process in a convective cloud system (described below), organic particles can adopt a highly porous morphology (Adler et al., 2013; Wagner et al., 2017). The porous morphology results from the fact that the initially liquid organic material concentrates and vitrifies upon ice formation, leaving behind a porous structure when the ice sublimates. This might provide a pathway to increase the ice nucleation ability of internally mixed AS/SOM particles with high organic mass fraction, because a porous organic coating might less efficiently cover the ice-active sites of the crystalline AS component. This pathway is investigated by our experiment of type E.

The freeze-drying process of the thickly SOM-coated AS particles was started at $-30\,°\mathrm{C}$ in the AIDA chamber. At this temperature, we can run through the droplet activation (liquefaction of the organic material) and freezing (re-vitrification of the organic material) within a single expansion cooling experiment. Figure 8a summarizes the cloud processing steps and the potential phase and morphology changes of the particles. The time series of the AIDA records during the freeze-drying expansion run, started at time $0$ s, are shown in Fig. 8b. When the relative humidity exceeded the water saturation (first vertical line in Fig. 8b), aerosol particles were activated as cloud droplets, detected by their larger diameter in the optical particle counter data. When the temperature approached the homogeneous freezing threshold ($\sim -36\,°\mathrm{C}$), a further fast drop in pressure was achieved by opening a valve in the pipe connection between the AIDA chamber and the evacuated APC chamber. This additional fast drop in pressure almost instantly reduced the gas temperature by another $2.5$ K and caused the entire droplet population to freeze homogeneously. The ice crystal formation is clearly visible in the optical particle counter data (second vertical line in Fig. 8b). The chamber was then refilled to ambient pressure with dry synthetic air and the ice crystals thereby quickly sublimated. To probe the ice nucleation ability of the freeze-dried aerosol particles at cirrus cloud conditions we further cooled the AIDA chamber to $-50\,°\mathrm{C}$. After the chamber cooling, the infrared spectrum of the freeze-dried AS/SOM particles in the regime of the $\nu_3$ ($\mathrm{SO_4^{2-}}$) mode was similar to spectra A and C in Fig. 7c, indicating that the

sulfate was effloresced. The AIDA expansion run started at $-50\,^{\circ}\text{C}$ and the INKA supersaturation scan was performed at $-54\,^{\circ}\text{C}$ with the particles sampled from the AIDA chamber at $-50\,^{\circ}\text{C}$. The results are shown in Figs. 8c and 8d with yellow symbols. As a reference, we also show the ice nucleating fraction (INF) curves for pure ammonium sulfate crystals (Exp. A1, orange symbols), and unprocessed, thickly SOM-coated AS crystals (Exp. B2, green symbols). The ice nucleation ability of the freeze-dried particles lies between those of the reference systems, similar to the in situ crystallized particles (experiment of type D). The heterogeneous ice nucleation mode of the freeze-dried particles has the ice onset for an ice nucleating fraction of $1\,\%$ at a saturation ratio $S_{\text{ice}}$ of $1.41$, while the ice onset saturation ratio is $1.33$ for pure AS crystals and $1.48$ for thickly SOM-coated AS crystals.

The activation curve of the freeze-dried particles measured with INKA (Fig. 8d, yellow data) has a different profile compared, for example, to the thickly-coated particles (green data). The slower increase of the ice nucleating fraction as a function of the ice saturation ratio (similar to the activation curve for AS in orange) suggests that heterogeneous freezing is the dominant ice nucleation mechanism occurring in INKA up to $S_{\text{ice}} \sim 1.6$. This effect could be related to the limited residence time of the particles in the INKA instrument, which together with a slower water uptake from the particles could have shifted the detected homogeneous freezing onset to higher $S_{\text{ice}}$ values. This kinetic limitation is not evident in the AIDA data, probably due to longer equilibration time during the expansion run. Particles in experiment E1 will likely be more viscous than particles from experiment B2 due to their larger organic mass fraction and/or due to a change of the chemical-physical properties of the particles after the freeze-drying process.

The enhanced ice nucleation ability of the freeze-dried particles may be explained by morphology changes. First, as already suggested above, a porous organic coating could less efficiently cover the ice nucleating active sites of the crystalline AS core. Second, the organic material might have been redistributed during the freeze-drying process, so that a similar morphology change as discussed in Exp. D could have occurred (like the formation of partially engulfed structures). The similar heterogeneous ice onsets measured in experiments D and E support this hypothesis. Third, the porous organic material formed during the freeze-drying process could be a better ice nucleus on its own via the pore condensation and freezing mechanism or by retaining an imprint of the sublimated ice on the highly viscous organic material. Previous studies observed an efficient preactivation of glassy organic aerosol particles after ice-cloud processing (Wagner et al., 2012; Kilchhofer et al., 2021). Wagner et al. (2012) investigated in the AIDA chamber four different organic solutes, raffinose, 4-hydroxy-3-methoxy-DL-mandelic acid (HMMA), levoglucosan, and a multicomponent mixture of raffinose with five dicarboxylic acids and ammonium sulfate, and observed that the ice-cloud processed glassy aerosol particles catalyzed ice formation at ice saturation ratios between $1.05$ and $1.30$. Kilchhofer et al. (2021) confirmed the improved ice nucleation ability of cirrus-cloud processed raffinose particles in dedicated CFDC ice nucleation measurements. This raises the question of why the ice-cloud processed AS/SOM particles (observed ice onset at $S_{\text{ice}} = 1.41$) do not reveal an even better ice nucleation ability. However, repeated ice nucleation experiments with secondary organic aerosol particles produced from the ozonolysis of $\alpha$-pinene performed by Wagner et al. (2017) showed a much lower susceptibility towards preactivation compared to the above mentioned organic solutes. For example, the ice nucleation onset of the ice-cloud processed $\alpha$-pinene SOA particles at $220\,\text{K}$ was at an ice saturation ratio of about $1.45$. A similar ice nucleation threshold was found in an experiment where highly porous $\alpha$-pinene SOA particles were formed by a

freeze-drying process at 243 K (as in the current study) and then cooled to cirrus temperatures to probe their ice nucleation ability. Furthermore, Adler et al. (2013) showed that in mixed AS/OM particles with a $1:1$ molar ratio, the formation of a porous structure was reduced compared to the pure organic particles, adding to the effect that the freeze-dried AS/SOM particles are not extremely efficient INPs.

In spite of the strong effect of thin SOM coating layers revealed by experiments of type C, we have shown that the ice nucleation ability of AS/SOM internally mixed particles is also strongly linked to their thermodynamic history. Uplifting of particles to the upper troposphere via convective clouds, for example, could lead to particles with an enhanced ice nucleation ability.

## 4  Conclusions

In this study, we investigated the ice nucleation ability of internally mixed particles composed of crystalline ammonium sulfate (AS) and secondary organic material (SOM) from the ozonolysis of $\alpha$-pinene. The ice nucleation ability of the particles was probed at temperatures between $-50\,^\circ$C and $-65\,^\circ$C, with expansion cooling experiments in the AIDA chamber and with a continuous flow diffusion chamber.

Mixed particles composed of sulfate and organics represent a major type of aerosol particles in the upper troposphere and may contribute a significant fraction of ice nucleating particles (INPs) involved in cirrus cloud formation (Froyd et al., 2009).

Crystalline AS particles are recognized efficient INPs at cirrus conditions (*e.g.*,  Abbatt et al., 2006). In this study we measured the ice nucleation onset of AS crystals at $S_{\text{ice}} = 1.29$ at $\sim -54\,^\circ$C, in agreement with literature results. However, we show that secondary organic material condensed on the AS surface progressively shifts the ice nucleation onset to higher ice saturation ratios. A small amount of SOM, corresponding to an organic mass concentration of $5-8$ wt%, is sufficient to increase the ice nucleation onset of the coated particles to $S_{\text{ice}} > 1.45$ (experiment of type C). Thus, a thin coating layer of secondary organic material is able to greatly reduce the ability of crystalline AS to act as INP. Möhler et al. (2008) also measured the suppression of the heterogeneous ice nucleating ability of mineral dust aerosol particles caused by a coating of SOM derived from the ozonolysis of $\alpha$-pinene. The ozonolysis of $\alpha$-pinene is not the only mechanism responsible for upper tropospheric SOM, as photolysis also plays a crucial role during daytime. However, the nucleation and growth mechanisms does not influence the ice nucleation ability of $\alpha$-pinene derived organic material (Piedehierro et al., 2021).

Internally mixed particles can form in the atmosphere through different pathways and can experience various aging and cloud-cycling processes, leading to particles with more complex morphologies than the idealized core-shell case, as simulated in experiments of type C. We performed two experiments to investigate whether there are pathways by which the ice nucleation ability of mixed AS/SOM particles with high organic weight fractions ($25-50$ wt%), as often found in the upper troposphere (Froyd et al., 2009), could be increased. Different particle morphologies, such as partially engulfed structures, might form, for example, when we start from homogeneously mixed, aqueous mixtures of sulfate and organics and then induce the crystallization of the AS component (experiments of type D). Our results show that the processed particles have an ice nucleation ability

lying between those of the pure and the SOM-coated AS particles in a core shell morphology, with the ice onset at $S_{ice} \sim 1.35$ at $-53\,°C$.

Another possible atmospheric process that can modify the ice nucleation ability of SOM-coated AS particles is the freeze-drying process. Previous studies have shown that highly viscous organic particles can adopt a porous morphology after a freeze-drying process, which could influence their ice nucleation ability. We simulated the freeze-drying process in the AIDA chamber at $-30\,°C$ and then probed the ice nucleation ability of the mixed processed particles at $\sim -50\,°C$ (experiment type E). Also in this case, the processed mixed particles show an intermediate ice nucleation ability compared to the reference systems (*i.e.*, ice nucleation ability between pure AS and AS with a compact SOM coating in a core shell morphology).

Our results from experiments D and E suggest that internally mixed particles, that undergo liquid or ice cloud processing, have unevenly distributed organic coating. The solid AS component is thus partially uncovered and able to catalyze the ice nucleation at lower $S_{ice}$ compared to the unprocessed, uniformly coated particles. In upcoming AIDA experiments, we aim at a better single-particle characterization to improve our knowledge about the morphology of the mixed AS/SOM particles, as already done in Wagner et al. (2017). However, this is not a trivial task, because the particles must be probed at low temperature in order to avoid potential morphology changes upon warming, where the reduced viscosity of the SOM component could lead to structural rearrangements.

In summary, the experiments presented in this study highlight the difficulty to represent the ice nucleation ability of internally mixed particles with just a single parametrization. The presence of an organic coating can suppress the ice nucleation ability of the seed particle at cirrus cloud conditions, but the ice nucleation ability of the same particle can substantially change if subjected to atmospheric processing. Since cloud processing is a common phenomenon, especially for particles uplifted in a convective system, their morphology changes and the enhancement of their ice nucleation ability need to be carefully characterized, also for internally mixed particles with different compositions (e.g., mineral dust particles coated with organic material).

*Data availability.* The data presented in this manuscript will be available through the KIT Open depository once the manuscript is accepted for publication.

*Author contributions.* BB and RW conceived and planned the experiments. BB, RW, JP, HS, and JS participated to the measurement campaign and operated the AIDA chamber. BB collected and analyzed the CFDC data. JS collected and analyzed the AMS data. RW collected and analyzed the FTIR data. BB prepared the plots and wrote the manuscript with major contributions from RW, KH, and OM. All authors contributed to the interpretation of the results and commented on the manuscript.

*Competing interests.* The authors declare that they have no conflict of interest.

*Acknowledgements.* We gratefully acknowledge the continuous support by all members of the Engineering and Infrastructure group of IMK-AAF, in particular by Olga Dombrowski, Rainer Buschbacher, Tomasz Chudy, Jens Nadolny, Steffen Vogt, and Georg Scheurig. This work has received funding by the European Union's Horizon 2020 research and innovation program under the Marie Skłodowska-Curie Grant Agreement 764991 and by the BMBF CLOUD-16 project (FZK $01LK1601C$). Additional funding has been received from the Helmholtz-Gemeinschaft Deutscher Forschungszentren.

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

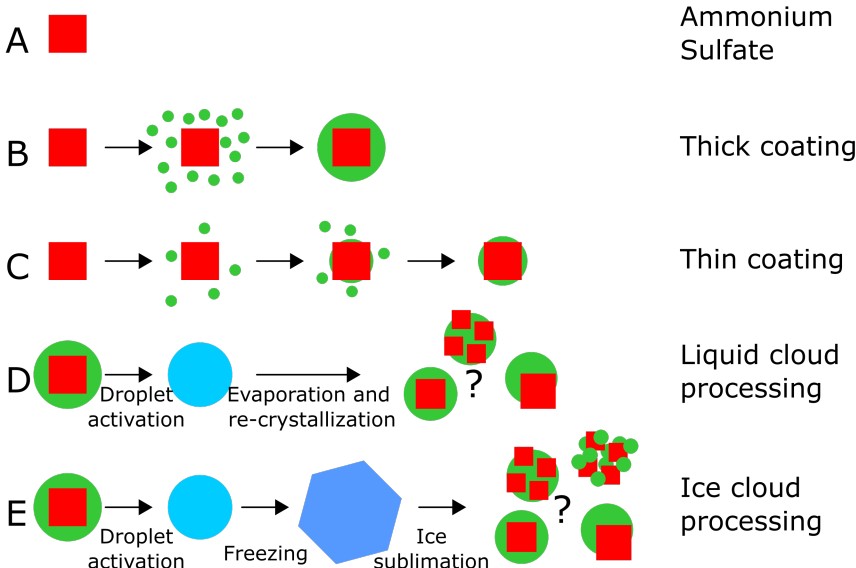

**Figure 1.** Summary of the experiments performed. Experiment **A**: crystalline ammonium sulfate (AS). Experiment **B**: AS crystals thickly coated with secondary organic matter (SOM) from the ozonolysis of $\alpha$-pinene (AP) at $+20\,^{\circ}\mathrm{C}$ in the aerosol preparation and characterization (APC) chamber (thick coating). Experiment **C**: AS crystals stepwise coated with small amounts of SOM from the ozonolyis of AP at upper tropospheric temperature in the AIDA chamber (thin coating). Experiment **D**: in situ crystallization of the AS component from a homogeneously-mixed, aqueous droplet that was generated by the droplet activation of crystalline AS aerosol thickly coated with SOM (prepared as in experiment B). Experiment **E**: aerosol prepared in the same way as in B, then subjected to an atmospheric freeze-drying process, involving droplet activation, freezing, and ice sublimation. A detailed description of the experiments is provided in the main text.

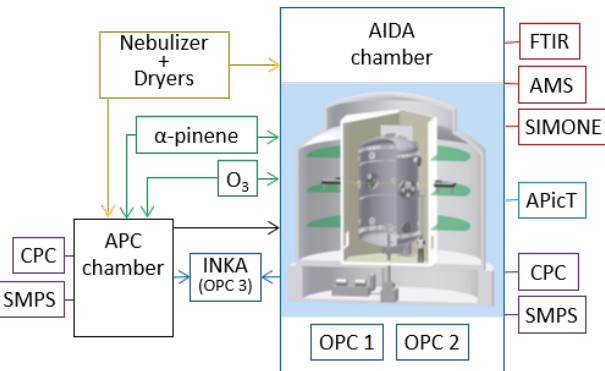

**Figure 2.** Schematic of the AIDA facility with experimental setup and instrumentation. **Yellow** outline: crystalline ammonium sulfate particles were generated with a nebulizer and a series of diffusion dryers. **Green** outlines: secondary organic matter was generated from the ozonolysis of $\alpha$-pinene. **Purple** outlines: particle number concentration and number size distribution measurements performed with condensation particle counters (CPC) and scanning mobility particle sizers (SMPS). **Light blue** outline: water vapor concentration was measured with an in situ multi-path tunable diode laser absorption spectrometer (APicT). **Red** outlines: the physico-chemical characterization of the aerosol particles was obtained from a Fourier transform infrared spectrometer (FTIR), a high resolution time of flight aerosol mass spectrometer (HR-ToF-AMS) and by light scattering and depolarization measurements (SIMONE instrument). **Blue** outlines: the ice nucleation ability of the investigated particles was measured with expansion cooling experiments in the AIDA chamber and with a continuous flow diffusion chamber (INKA). OPC: optical particle counter. More details are provided in the main text.

**Table 1.** Summary of the experiments performed and main parameters of the aerosol particle populations. $D_p^{\mathrm{median}}$ is the median diameter of the lognormal fit to the aerosol size distribution measured right before the coating procedure. The $\alpha$-pinene concentration corresponds to the total amount added during the injection periods. For the in situ coating experiments (experiments type C), the organic mass fraction $f_{\mathrm{org}}$ refers to the value measured at the end of the coating procedure. The coating thickness is estimated from a combined analysis of the SMPS and the AMS data, the first number refers to the coating thickness evaluated at the median diameter, and the range in the parenthesis to the values at 300 and 500 nm particle diameter (see text for more details).

| Exp. ID | Type | $D_p^{\mathrm{median}\,*}$ [nm] | Coating T [°C] | Coating RH [%] | $O_3$ [ppm] | $\alpha$-Pinene [ppb] | Coating time | $f_{\mathrm{org}} \cdot 100$ [wt%] | Coating thickness [nm] |
|---------|------|------|------|------|------|------|------|------|------|
| A1 | Amm. Sulfate | 395 | - | - | - | - | - | - | - |
| B1 | Thick coating | 380 | +25 | < 10 | 1.6 | 110.9 | 33 min | 26.9 | $28\,(22-37)$ |
| B2 | Thick coating | 415 | +25 | < 10 | 2.6 | 325.1 | 1 h 4 min | 39.1 | $50\,(36-60)$ |
| C1 | Thin coating | 403 | −50 | 60 | 0.2 | 0.6 | 4 h 10 m | 5.7 | $6\,(4-7)$ |
| C2 | Thin coating | 430 | −60 | 57 | 0.2 | 1.2 | 2 h 45 min | 4.8 | $5\,(3-6)$ |
| C3 | Thin coating | 403 | −45 | 61 | 0.2 | 0.7 | 5 h 10 min | 8.3 | $8\,(6-10)$ |
| D1 | Liquid cloud | 380 | +25 | < 10 | 1.8 | 238.7 | 1 h | 39.1 | $46\,(36-60)$ |
| D2 | Liquid cloud | 365 | +25 | < 10 | 1.6 | 110.4 | 33 min | 24.3 | $24\,(20-33)$ |
| E1 | Freeze-drying | 345 | +25 | < 10 | 1.8 | 235.1 | 1 h 1 min | 47.3 | $54\,(47-79)$ |

$^{*}$ Assuming a shape factor $\chi = 1.1$

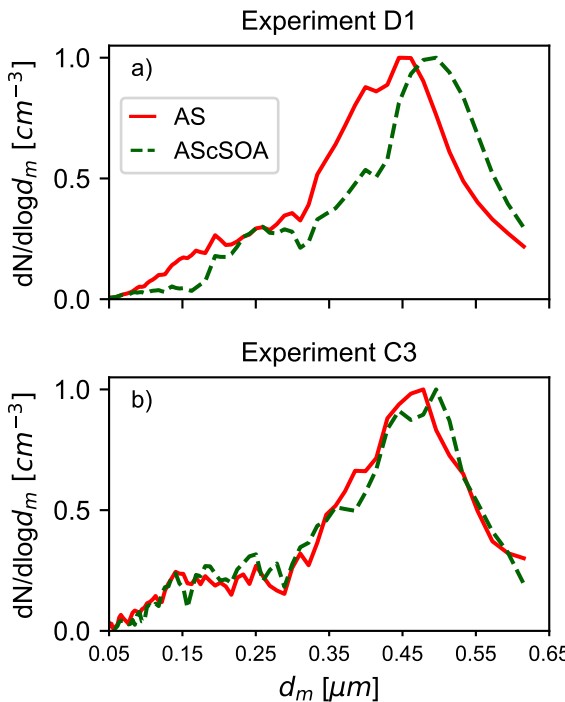

**Figure 3.** Comparison of the normalized particle number size distribution before and after the coating procedure. The x-axes represent the electrical mobility diameter $d_m$. **(a)** Particle number size distribution for experiment D1 before (red) and after (green) the coating process. The thick organic coating was performed in the APC chamber. **(b)** Particle number size distribution for experiment C3 before (red) and at the end (green) of the in situ coating procedure. The thin coating experiments were performed stepwise and with lower concentrations of the gas precursor in the AIDA chamber. No apparent shift in the particle number size distribution was detected for experiments of type C.

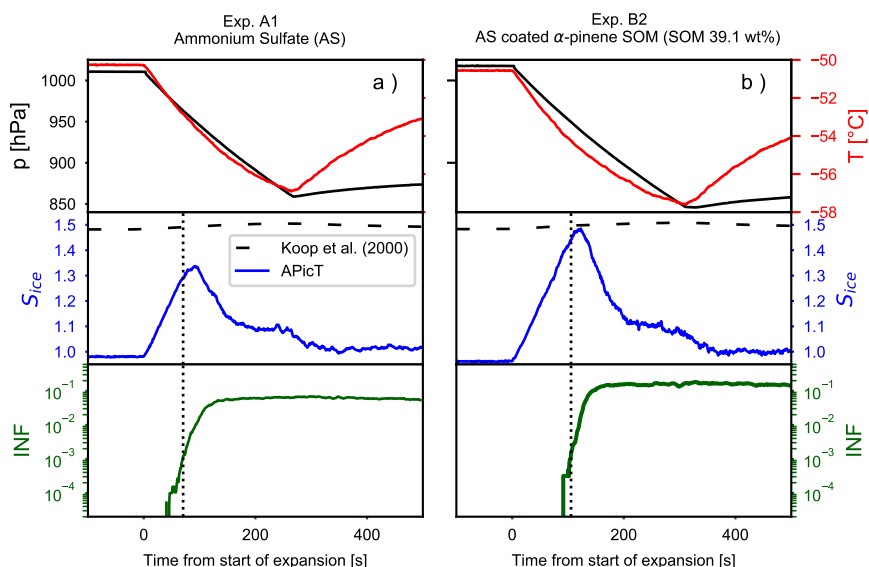

**Figure 4.** Time series of the AIDA expansion cooling experiments for **a)** pure crystalline ammonium sulfate (experiment A1) and **b)** ammonium sulfate crystals coated with SOM from the ozonolysis of $\alpha$-pinene (experiment B2). The upper panels show pressure (black lines, left axis) and temperature (red lines, right axis). The middle panels show the increase of the saturation ratio with respect to ice (blue line). The homogeneous freezing threshold computed for a $\Delta a_{\mathrm{w}}$ of $0.3$ of the ice-melting curve (Koop et al., 2000) is indicated with the horizontal dashed lines. The lower panels show the fraction of aerosol particles that induced ice formation (ice nucleating fraction, INF). The vertical dotted lines indicate when $0.1\,\%$ of the aerosol particles acted as INPs.

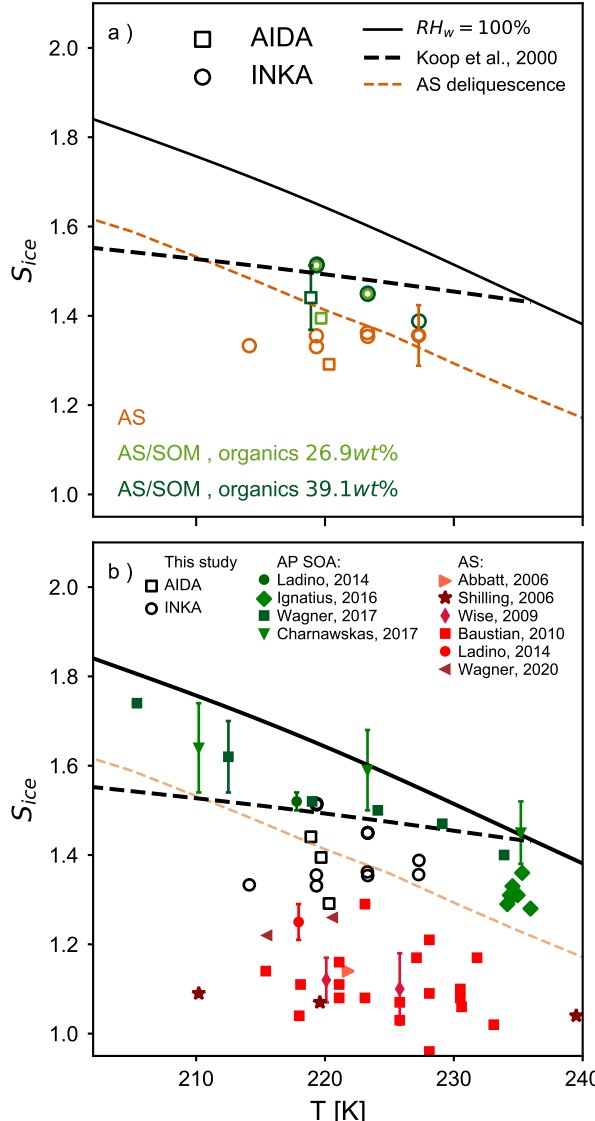

**Figure 5.** Summary of the ice nucleation onsets for crystalline ammonium sulfate (AS), pure secondary organic aerosol (SOA) particles and, crystalline AS coated with secondary organic matter as a function of temperature T. The solid black line indicates water saturation ($RH_w = 100$ %), the dashed black line indicates the homogeneous freezing threshold ($\Delta a_w = 0.3$, Koop et al. (2000)), and the dashed orange line is the ammonium sulfate deliquescence relative humidity (Clegg et al., 1998). **a)** Results from this study (corresponding to an ice nucleating fraction of 0.1 %) for pure, crystalline AS (orange symbols) and for AS coated with $\alpha$-pinene SOM (green symbols). Light green symbols refer to experiment B1 (organic mass fraction 26.9 wt%), dark green symbols refer to experiment B2 (organic mass fraction 39.1 wt%). Results from the AIDA expansion cooling experiments are reported with squares, circles refer to INKA results. **b)** Results from this study (open symbols) are compared to literature data for crystalline AS (red symbols) and $\alpha$-pinene SOA particles (green symbols).

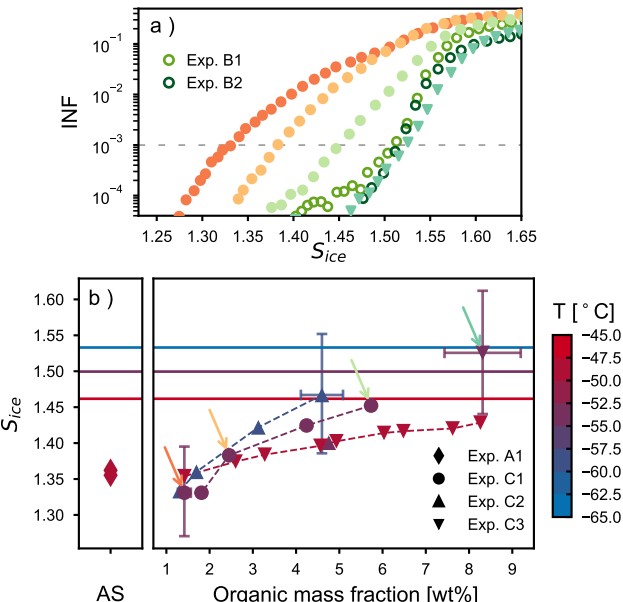

**Figure 6.** Ice nucleation results for the thin coating experiments (type C). **a)** Ice nucleating fraction as function of $S_{ice}$ for humidity scans performed with INKA at $\sim -54\ ^\circ\mathrm{C}$ during experiments C1 (solid circles), C3 (down-pointing triangles), and B (empty circles). **b)** Ice onsets saturation ratio (defined for an INF of $0.1\%$) for all INKA measurements during experiments C1, C2, and C3. Colors indicate the temperature at which the ice nucleation measurements have been performed in INKA. The horizontal lines indicate the threshold for homogeneous freezing of solution droplets at three different temperatures ($\Delta a_{\mathrm{w}} = 0.3$, Koop et al. (2000)). In all experiments, a gradual increase in the ice onset saturation ratios with increasing organic mass fraction was measured (dashed lines to guide the eyes). The colored arrows indicate the measurements reported in panel a.

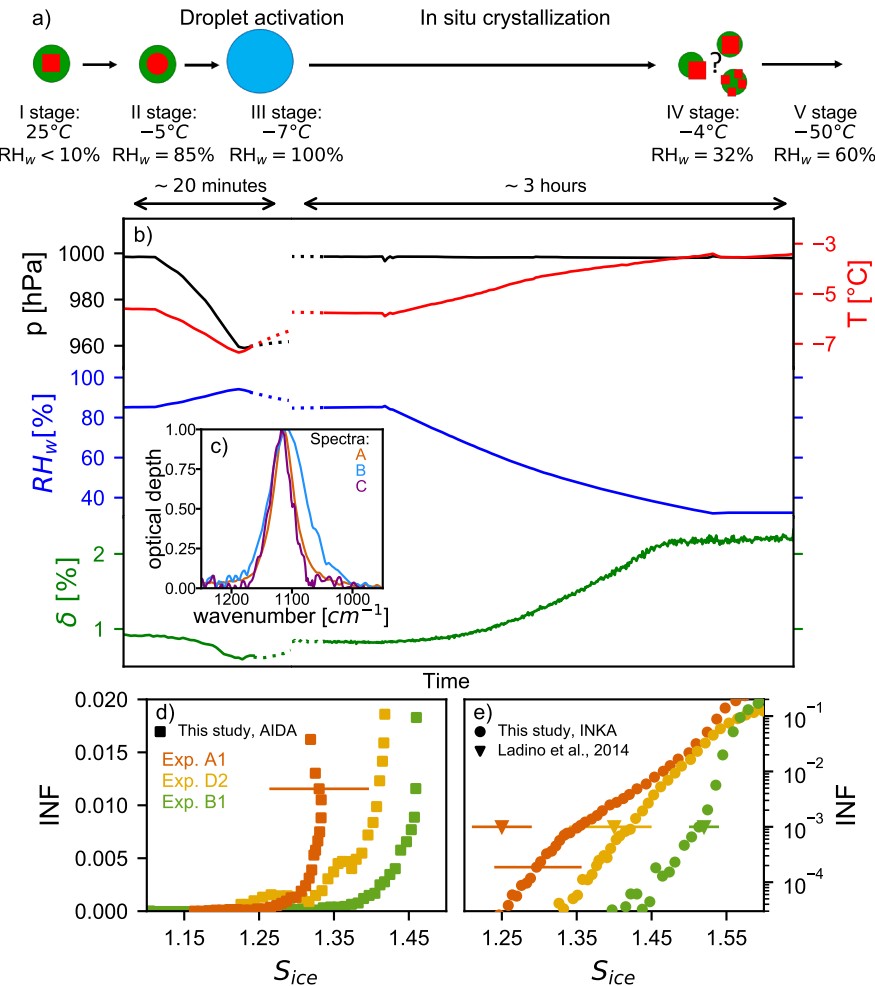

**Figure 7.** Liquid cloud processing, re-crystallization, and ice nucleation results (experiment type D). **a)** Schematic of the phase state and possible morphology of the particles after the in situ crystallization process. **b)** Time series of pressure (black line), temperature (red line), relative humidity with respect to water (blue line), and backscattering linear depolarization ratio (green line) during the in situ crystallization for experiment D2. The left part of the plot shows the short expansion run started at $-5\,^{\circ}$C and $RH_w = 85\,\%$ to activate the seed aerosol particles to homogeneously mixed, aqueous droplets. After refilling of the AIDA chamber to ambient pressure (not shown in the figure), the relative humidity was reduced to $RH_w = 32\,\%$ to induce the AS efflorescence. **c)** Normalized infrared spectra in the regime of the $\nu_3$ ($SO_4^{2-}$) mode obtained after transfer of the particles into the AIDA chamber (spectrum B) and after the in situ crystallization (spectrum C) in comparison with a reference spectrum of pure AS crystals (spectrum A). **d)** and **e)** Ice nucleating fraction as function of $S_{ice}$ for the AIDA and INKA experiments with pure AS crystals (experiment A1, orange symbols), unprocessed AS crystals with thick SOM coating (experiment B1, green symbols), and in situ crystallized mixed AS/SOM particles (experiment D2, yellow symbols). Ice onset results from Ladino et al. (2014) are reported in panel e as triangles: pure AS in orange, mixed AS/SOM particles in yellow, and pure SOA particles in green.

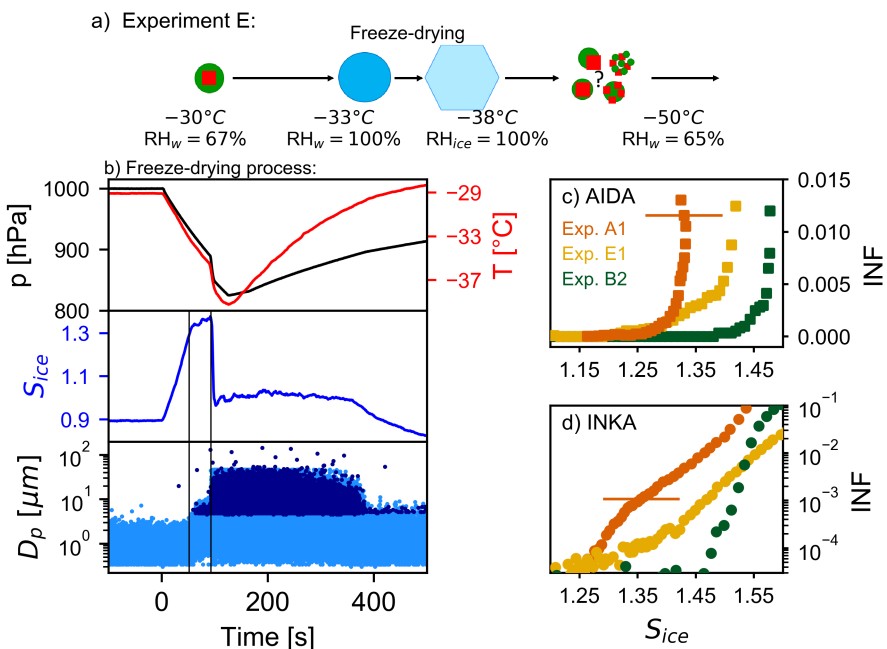

**Figure 8.** Results for the freeze-drying experiment (experiment E1). **a)** Scheme of the freeze-drying process together with the possible phase state and morphology of the particles during the experiment. **b)** Time series of the AIDA data during the freeze-drying process. Upper panel: pressure and temperature inside the AIDA chamber (black and red traces). Middle panel: saturation ratio with respect to ice (blue trace). Lower panel: single particle data from the welas optical particle counters (OPC1 and OPC 2), each dot corresponds to an aerosol particle, droplet, or ice crystal at the corresponding size. **c)** and **d)** Ice nucleating fraction as function of the saturation ratio with respect to ice measured by AIDA and INKA. Data for pure AS crystals are reported in orange (experiment A1), unprocessed AS crystals with thick SOM coating in green (experiment B2), and freeze-dried SOM-coated AS crystals in yellow (experiment E1). Note the different scale in panels c and d.