# Peer review of "Ice nucleation ability of ammonium sulfate aerosol particles internally mixed with secondary organics"

_Atmospheric Chemistry and Physics, 2021_

## Referee Comment (RC1)

**Anonymous Review of Ice nucleation ability of Ammonium Sulfate aerosol particles internally mixed with Secondary Organics**

Anonymous Reviewer

March 10, 2021

**1 Summary**

In this work, Bertozzi et al. investigated the heterogeneous ice nucleation ability of ammonium sulfate (AS) coated with $\alpha$-pinene SOA (dark ozonolysis) from -65 to -50 °C. These mixed particles were generated to have a core-shell morphology, or to change their morphology through droplet activation and drying or homogeneous ice nucleation and subsequent sublimation. Ice nucleation by these particles was measured by the AIDA chamber, and also the continuous-flow diffusion chamber INKA. Both of instruments are well known and well characterized, and their operation and limitations are well represented in this paper.

This manuscript consisted of four different experiments. Ice nucleation on bare AS, AS with thick and thin SOA coatings (AS+SOA), AS+SOA particles that underwent droplet activated and subsequent evaporation below the AS deliquescence point, and AS+SOA particles that underwent homogeneous ice nucleation and subsequent sublimation. These experiments were well planned and well thought out. The last two experiments were novel, and take advantage of the unique capabilities of combining the AIDA chamber with INKA.

In this work, the authors found that their ice nucleation onsets for ammonium sulfate and thickly coated AS+SOA particles agreed with previous literature values, with bare AS having an ice nucleation onset of $S_{ice} = 1.3$, and the SOA coatings inhibiting ice nucleation until $S_{ice}$ was $\geq 1.45$. They also found that organic mass fractions of 5-8% were enough to cause this shift in $S_{ice}$ onset. Finally, the authors found that the droplet activated/evaporation and ice formation/sublimation changed the $S_{ice}$ onsets for even thickly coated AS+SOA. For these particles, the onsets were in between bare AS and thickly coated AS+SOA in a core-shell configuration, or an $S_{ice}$ of 1.35.

The paper was a refreshing read. It was well written and well organized. Overall, I have only two major comments, and a few minor/technical comments.

**2  Major Comments**

1. On P7L212, you use the median diameter to derive your coating thickness. How good of an assumption is this? For example, how does the mass fraction of organics change with size for a single experiment, and how does the coating thickness change a function of size for a given mass fraction? You should be able to derive these quantities from the SMPS and AMS.

2. On P15L490, you start discussing the $S_{ice}$ onset for the freeze-dried particles. I was surprised to see that the particles did not nucleate ice extremely efficiently as suggested in the paper by Adler et al, 2013. This suggests that the AS+SOA particles were not glassy, or had sufficient self-diffusion rates to not retain a perfect imprint of the sublimated ice. This needs to be discussed further in the paper. You should be able to tell from the IR if the sulfate was locked in glassy matrix or effloresced, correct?

**3  Minor/Technical Comments**

- Title: I believe that "Ammonium, Sulfate, Secondary, and Organics" should all be lowercase.

- P2L38: To me, the phase "allows to" here reads very awkward. Perhaps change to "allows us to?"

- Equation 1: What was the prescribed density of organics in this work?

- P8L249: Here the phrase "allow to" shows up again and reads slightly awkward. I would suggest deleting it and changing "evacuate" to "evacuates."

- P9L274: Perhaps change "thanks to" to "because of."

- P9L285: Perhaps change "allows to minimize" to "minimizes"

- P10L305: Change "that" to "which."

- P11L340: Perhaps change "reddish" to "red." I see that the symbols are slightly different in color, but, in this reviewer's opinion, it's okay to simple say "red."

- P11L341: Perhaps change "greenish" to "green."

- P13L415: The sentence "Note that in this specific experiment the chamber walls were not coated with ice" could be deleted. It feels extraneous to this reviewer.

- P14L434: Could cite the work of Freedman et al., 2020 here.

- P15L468: It seems as if the authors are suggested that the coated particles were not heterogeneously ice active, when Figure 5 seems to suggest they are. Perhaps the authors mean that they have higher onsets than homogeneous nucleation?

- P16L518: Same comment as the one previous–the authors are implying that thickly coated particles are not "active as INPs"

---

## Referee Comment (RC2)

**Review of Manuscript acp-2021-53:**
**Ice nucleation ability of Ammonium Sulfate aerosol particles internally mixed with Secondary Organics**
**Barbara Bertozzi, Robert Wagner, Kristina Höhler, Joschka Pfeifer, Harald Saathoff, Junwei Song, Thomas Leisner, and Ottmar Möhler**

Bertozzi et al present a very interesting and worthwhile study on the cirrus temperature regime ice nucleation abilities of crystalline ammonium sulphate and secondary organic material derived from the dark reaction products of $\alpha$-pinene ozonolysis. The experiments are carefully conducted with interesting and valuable conclusions that would be of significant interest to the ice nucleation community. I enjoyed reading this contribution and commend the authors on the experiments. I recommend publication after the following comments are addressed.

Major points:

The experiments conducted here in are done systematically with 5 different systems investigated (Experiments A – E). I think the discussion of cloud processing and its impacts have been kept too implicit. I think the authors could do a better job in addressing how cloud processing of organic coated AS can enhance the ice nucleation activity compared to unprocessed particles. The discussion is often presented from the other point of view, i.e. compared to pure crystalline AS, the cloud processed particles are less IN active, however, given the cloud cycling is a commonly occurring (REF) process (both through liquid and ice clouds) in the atmosphere, I recommend also making the comparison and bringing out the cloud processing aspect of the study more. Also, in the atmosphere more often than not, particles are internally mixed, as such their cloud processing studies would be most relevant for the troposphere.

Can the authors better justify their definitive claim that the particles experimented with had core-shell morphology in the paper, even in the case of the thick coatings in Experiment B. To me it seems like this is being taken for granted, but there is no definitive evidence being provided. Especially in Experiment B where the coatings are conducted at room temperature and low RH, glassy state for the SOM is favoured and thus how likely is it to form core shell morphology.

While reading the paper, I think there is some evidence to possibly infer the presence of core-shell morphology, but the way the paper is written now, it is assumed from the get-go, that indeed this is the case, without leaving room doubt. I think at best the authors claim they believe it is core-shell morphology, or it should be clearly stated how they can definitively support this claim. I make suggestions below, of how such a claim could be inferred based on their ice nucleation results.

**Specific comments and suggestions:**

Line 49: Change "accordance" to "agreement"

Line 59: Oxalic acid can also act as an INP in the deposition mode [*Kanji et al.,* 2008]

Line 67: could add *Möhler et al.* [2008] here as a reference for homogeneous freezing of pure SOM

Line 68: could add references *Möhler et al.* [2008] and *Kanji et al.* [2019] as examples of studies that used the same SOM as proxies for organic particles and coatings

Line 106: it maybe more accurate to say, "ice nucleation active" or "not ice nucleation active". Ice inactive is a little too colloquial and can have broader meaning.

Line 108: Delete comma after "humidity"

Line 121: consider using "ice nucleation active" instead of "ice-active"

Line 122-126: here the authors describe Experiment D and refer to it as also forming liquid droplets, then why not just call this process liquid cloud processing. This would be more consistent with the terminology used in Experiment E. It is after all a process where liquid drops are formed first, followed by evaporation. So liquid cloud processing would be more suitable. Afterall the only difference between D and E is the absence of ice cloud formation. See comments on Figure 1.

Line 176-177: Should read "The coating procedure in the APC chamber was performed at ambient temperature and low relative humidity and in the AIDA chamber at low temperature and ice saturated conditions."

Line 183: Can the RH in the injection line be better quantified. For injection into AIDA 10% $RH_w$ at room temperature could be very high for the cold temperatures conducted below 233 K. Thus, this could imply that the particles deliquesced upon entering the AIDA chamber. Please clarify this point in the manuscript.

Line 197: Delete the word "again"

Line 222-223 where the coating thicknesses are given, can the $RH_w$ be given as well, to have an impression of the phase state of the SOM. The temperature is given in Table 1. Ideally the $RH_w$ could also be included in Table 1.

Line 237-238: should read "In the backward direction, the polarization-resolved scatter light intensity is detected so that …"

Line 256: should read "The nucleated ice quickly grows to large crystals and…"

Lines 281-285: What is the temperature of the evaporation section set at. Is it set at the sample temperature, cold wall or warm wall temperature? This should be mentioned followed by the justification that the authors correctly address that the evaporation section will not influence the measurements in this study since it is conducted at cirrus conditions and any further ice nucleation in the evaporation section would require $RH_i$ to be larger than 100% which is not the case in the evaporation section. However, in the case liquid drops form in the growth section and if the evaporation section is at the cold wall

temperature, then the drops could freeze before evaporating. This should not be of issue in the current study because of the low temperatures investigated, but should be clarified in terms of how the INKA chamber operates.

Line 311: the core-shell morphology at this point is stated with certainty. How can this be verified? Either provide more evidence that this is indeed the case, or the authors can suggest at this point that "we believe it to be core-shell morphology". See further below for how this can be inferred based on the arguments the authors already provide.

Line 314-315: the thicknesses are calculated assuming uniform/even coating. This should be added to the end of the sentence. E.g. "…thickness *of* 44 nm assuming an even/uniform coating".

Line 332: replace "notified" with "noted"

Line 335-339: one may want to simply state that longer equilibration times are possible in the AIDA chamber compared to the CFDC, which is sort of stated, but not explicitly.

Lines 358-365: What is shown in Figure 6a, that cannot be deduced from Figure 6b. The time perhaps, but this doesn't come up in any discussions or an anchor point for an argument. It looks like figure 6a is redundant and figure 6b nicely shows the point the authors are trying to make and with more clarity. I suggest removing Fig. 6a or make clearer what additionally it contributes to the discussion.

Line 368-369: The measurements definitely demonstrate the possibilities of combining CFDC with ageing experiments, but I don't think the statement about combining AIDA with CFDC is needed as mentioned in this sentence. AIDA is a complex chamber with highly sophisticated instruments coupled to it and capabilities. For the experiments being proposed here a chamber that can be cooled to low enough temperatures can be used here coupled with a CFDC. i.e. this is in reference to the step wise coating experiments in Experiment C. A chamber as complex and sophisticated as AIDA would not be necessary.

Thank you for this remark. We have removed that sentence.

Line 370 – 377: Could the authors also comment on the onset $S_i$ in Fig 6c for experiment C1-C3 at the same organic coating % i.e. why is the onset $S_i$ lower for the warmer T experiments than the colder T experiments at the same organic coating %? From ice nucleation perspective the ice germ should be smaller at colder T, so the onset for colder T should be at lower $S_i$ or the same. Is it a diffusional growth limitation to when the OPC detects ice crystals or is it a limitation of diffusing water vapour into the organic coating? Some comment addressing the difference in onset $S_i$ for the same organic coating percent as a function of temperature should be made here.

Line 421: should read "….with INKA at -54 °C immediately…"
Line 422: delete "at -54 °C"

Line 424: having the diffusion dryer upstream of the INKA instrument, could have modified the phase state of the organic from liquid to glassy, as such would it not be correct to state

that INKA is already sampling glassy or phase separated aerosol since the LLPS occurs at $RH_w$ < 90%.

Line 423-434: This argument presented here is quite interesting and can be exploited to support the authors' claim that core-shell morphology or complete coating was achieved in the thick coating experiments (B). The reason being that Experiment B completely suppressed the het. ice nucleation signal compared to that of pure AS in experiment A. Then coatings as thick as those in Experiment B but in Experiment D (liquid cloud processing) resulted in intermediate IN activity. As the authors point out that this must have resulted in changes in morphology allowing the initially thickly coated particles to become ice nucleation active after the liquid processing. This observation together with Experiment B and A would then more concretely support the core-shell morphology of the thick coatings. I think before this point, it is hard to lay claim to it so definitively without more concrete evidence.

Line 451: I think the authors should specify that they mean > 5% by coating mass when they say "even thin coatings" because below 5% there was ice nucleation activity below the homogeneous freezing threshold suggesting that coatings with mass % < 5% will not completely suppress the het. IN activity.

Line 456: why would the crystalline AS core not dissolve. It was not clear to me how immersion freezing by the crystalline core is being explained here. Shouldn't the AS core dissolve in water if it is immersed?

Line 461-462: Why would AS efflorescence be inhibited at low temperatures, can this be explained more clearly to make the connection to the point above this sentence.

Line 479: The authors should be careful in describing the freeze-drying process. After activation into drops and freezing, just completes the freezing part of the freeze drying, and then upon sublimation comes the drying part, as such saying that all aerosol particles underwent freeze-drying in line 479 is still too early. It is only after line 481 where the authors state "…thereby quickly sublimated" is when the freeze-drying process has been accomplished.

Line 499. Here one more discussion point could be added is the how cloud processing enhances ice nucleation of organic coated particles. This is not brought out explicitly enough. See major comment 1.

Line 505 and elsewhere where troposphere is mentioned. A mention that the SOM here may not be representative of tropospheric SOM is warranted because almost always photochemistry would play a role in the nature of SOM in the atmosphere, and here no UV source is mentioned in the production of SOM. This should be acknowledged somewhere in the manuscript. I suggest once in the methods where the SOM production is described and then again in the conclusion section.

Line 510-513: this part ignores the fact that coating mass % below 5% still yield het. ice nucleation active particles. This should be acknowledged here.

Line 527: should read "(i.e., ice nucleation ability between pure AS and AS with a compact…)"

**Figures and Tables**

Figure 1: Why is experiment D not called liquid cloud processing, to be consistent with Experiment E which can be called cold cloud or ice cloud processing. The only difference between the two is freezing followed by sublimation vs. droplet activation followed by evaporation. Alternatively, D could be called liquid cloud processing and E could be called cloud processing since E involves both liquid and ice.  I would consider making the labelling more consistent. Even in the figure caption the authors state in the last line that E involves droplet activation thus it demonstrates the similarity between D and E (thus I would suggest both be part of a cloud processing label).

In the figure caption "upper tropospheric conditions" is mentioned, but I think to be more specific the authors should say upper tropospheric temperature, since photolysis is not part of this work, or UV is not been simulated in this study which would be part of UT conditions.

Table 1: $O_3$ concentration can be changed to ppm and scientific notation removed and all numbers harmonised in terms of how many significant digits are given. Significant digits can also be harmonised for all experiments of $\alpha$-pinene concentrations.

Figure 3. Would be helpful for reading if more than 3 ticks are added to the x-axis of panels a and b. i.e. every 0.1 μm and also add the scale on the x-axis of panel a.

Figure 5: panel a, either label all experiments as Exp A/B1/B2 or then AS, AS/SOM thick coating etc. but currently it is switched from AS then EXP B1/B2. Choose one style to be consistent.

Panel b. the contrast between the literature results in red and the orange data from the current study isn't that great. Also, the distinction between the green from the current study and literature is not that clear either. To make this clearer, I would suggest keeping the data from this study in black since AIDA and INKA are already shape differentiated. This will make it very clear to a viewer of where and how the data from this study compare to the literature.

Figure 6: As mentioned in the text above, the purpose of panel a is not clear to me. It could be removed. Also, in the text of the manuscript one could describe that experiment C nicely shows that gradually increasing the coating thickness progressively suppresses the IN activity. So far it is discussed as though this happens as a step function at 5%, but the experiments nicely show that this is a gradual process of $S_i$ increasing with organic mass fraction.

Figure 7. One could consider another panel, where the activation curves for Exp A1, and 3 mass fractions of C2 and B1 are plotted, to show the progressive nature and that experiments with coatings larger than 5% yield the same result as that of ~30%.

Figure 8. what are the different blue coloured dots in panel b? This is not described in the caption. Also, the green curve crossing the yellow curve in panel D is not discussed or addressed anywhere in the text. This crossing suggests that at a certain saturation the thickly coated particles are more active than the cloud processed particles (even if it is in the homogeneous freezing regime).

**References**

Kanji, Z. A., O. Florea, and J. P. D. Abbatt (2008), Ice formation via deposition nucleation on mineral dust and organics: dependence of onset relative humidity on total particulate surface area, *Env. Res. Lett.*, *3*(2), doi:10.1088/1748-9326/3/2/025004.

Kanji, Z. A., R. C. Sullivan, M. Niemand, P. J. DeMott, A. J. Prenni, C. Chou, H. Saathoff, and O. Möhler (2019), Heterogeneous ice nucleation properties of natural desert dust particles coated with a surrogate of secondary organic aerosol, *Atmos. Chem. Phys.*, *19*(7), 5091-5110, doi:10.5194/acp-19-5091-2019.

Möhler, O., S. Benz, H. Saathoff, M. Schnaiter, R. Wagner, J. Schneider, S. Walter, V. Ebert, and S. Wagner (2008), The effect of organic coating on the heterogeneous ice nucleation efficiency of mineral dust aerosols, *Env. Res. Lett*, *3*(2), 025007, doi:10.1088/1748-9326/3/2/025007.

---

## Author Comment (AC1)

We thank the anonymous Reviewer for the interesting and useful comments, which help to improve the quality of our article.

Our replies are reported in the following with blue fonts, Reviewer's comments are reported in black. Reporting the changes made in the manuscript, we indicate with strikethrough fonts the text previously present in the manuscript and now removed, and with bold fonts the additions we suggest to include in the revised manuscript. References to pages and lines in the responses refer to the revised version of the manuscript, unless noted otherwise.

**Major Comments**

1. On P7L212, you use the median diameter to derive your coating thickness. How good of an assumption is this? For example, how does the mass fraction of organics change with size for a single experiment, and how does the coating thickness change a function of size for a given mass fraction? You should be able to derive these quantities from the SMPS and AMS.

Thanks for this interesting comment. We have looked at the size-resolved particles' chemical composition as measured by the AMS. In Fig. A we show the results for two exemplary experiments: B1 (coating performed in the APC chamber) and C3 (coating performed in the AIDA chamber). The normalized mass concentration of organics (green), sulfate (red), and ammonium (orange) are shown as a function of the vacuum aerodynamic diameter $D_{va}$. The three curves overlap nicely for particles with vacuum aerodynamic diameter equal to or larger than the main mode in both experiments, meaning that the organic mass fraction $f_{org}$ is not size-dependent. The slightly higher organic mass fraction for smaller particles can be related to some pure SOA particles nucleated during the coating process, or a higher mass fraction condensed on the smaller AS seeds.

[Figure]

*Figure A:* Normalized mass distributions measured by the AMS after coating for experiments B1 and C3 as a function of the aerodynamic vacuum diameter $D_{va}$. Organics are shown in green, sulfate in red, and ammonium in orange.

As suggested by Referee #1, we then evaluated the coating thickness $d$ as a function of the seed particle size $D_p$ for a given organic mass fraction $f_{org}$. The equations and the text in the manuscript have been adapted accordingly.

From line 217:
"The organic mass fraction values $f_{org}$ reported in Table 1 refer to the organic content at the end of the coating procedures, i.e., regarding experiments of type C, they refer to the organic content after the last coating step. **The size-resolved measurements of the chemical composition of the particles by the AMS indicate that the organic mass fraction is constant and not a function of the particle size.**
To estimate the thickness of the organic coating, d, we combined the size distribution of the pure crystalline AS particles from the SMPS measurements and the organic mass  **fraction** $f_{org}$ from the AMS. Thereby, we assume the crystalline AS seeds to be  spherical particles with diameter $D_p$ , **and** the organic material  **to be** evenly distributed on  **their** surface, leading to a spherical organic shell. The coating thickness d is thus calculated assuming a perfectly concentric core shell morphology as follows:

$$\cancel{D_p^{coated} = (\frac{6}{\pi}\frac{M_{org}}{C_n \cdot \rho_{org}}+D_p^3)^{\frac{1}{3}}}$$

$$d = \frac{D_p^{coated} - D_p}{2}$$

 **The** diameter of the  **coated particles $D_p^{coated}$ is calculated by considering the size-dependence of the particles' organic mass $M_{org}(D_p)$:**

$$D_p^{coated}(D_p) = [D_p^3 + \frac{6}{\pi}\frac{M_{org}(D_p)}{\rho_{org}}]^{\frac{1}{3}}$$

**With:**

$$M_{org}(D_p) = \frac{f_{org}}{1 - f_{org}} \cdot \rho_{AS} \cdot \frac{\pi}{6}D_p^3$$

The resulting estimated organic coating thickness  **corresponding to the median diameter of the particle population** is reported in Table 1. **In Table 1, we also indicate between brackets the estimated coating thickness for seed particles with diameters of 300 and 500 nm, i.e., the size range which comprises the major particle mode in the number size distribution (see below and Fig. 3 for the number size distribution measurements).** The thickness estimated for the thickly SOM-coated AS particles (experiments of type B, D and E) ranged from  **24 nm to 54 nm**. Thinner coatings were obtained at the end of the coating experiments performed in the AIDA chamber (experiments of type C) with  **6 nm**, 5 nm, and  **8 nm** coating thicknesses."

2. On P15L490, you start discussing the Sice onset for the freeze-dried particles. I was surprised to see that the particles did not nucleate ice extremely efficiently as suggested in the paper by Adler et al, 2013. This suggests that the AS+SOA particles were not glassy, or had sufficient selfdiffusion rates to not retain a perfect imprint of the sublimated ice. This needs to be discussed further in the paper. You should be able to tell from the IR if the sulfate was locked in glassy matrix or effloresced, correct?

- Thank you for raising this interesting aspect that certainly warrants further discussion in the manuscript. There are a couple of previous studies that observed an efficient preactivation of organic aerosol particles after ice-cloud processing, which may be explained by the formation of a porous structure or the retention of an imprint of the sublimated ice on the highly viscous organic material. Wagner et al. (2012) investigated four different organic solutes, raffinose, 4-hydroxy-3-methoxy-DL-mandelic acid (HMMA), levoglucosan, and a multicomponent mixture of raffinose with five dicarboxylic acids and ammonium sulfate, and observed that the ice-cloud processed glassy aerosol particles catalyzed ice formation at ice saturation ratios between 1.05 and 1.30. These experiments were also conducted in the AIDA chamber. Recently, Kilchhofer et al. (2021) confirmed the improved ice nucleation ability of cirrus-cloud processed raffinose particles in dedicated CFDC ice nucleation measurements. However, repeated ice nucleation experiments with secondary organic aerosol particles produced from the ozonolysis of α-pinene performed by Wagner et al. (2017) in the AIDA chamber showed a much lower susceptibility towards preactivation compared to the above mentioned organic solutes. For example, the ice nucleation onset of the ice-cloud processed α-pinene SOA particles at 220 K was at an ice saturation ratio of about 1.45. A similar ice nucleation threshold was found in an experiment where highly porous α-pinene SOA particles were formed by a freeze-drying process at 243 K (as in the current study) and then cooled to cirrus temperatures to probe their ice nucleation ability.
  So far, there is therefore no experimental evidence that the freeze-dried α-pinene SOA particles must necessarily be extremely efficient INPs. Furthermore, Adler et al. (2013) showed that in mixed AS/OM particles with a 1:1 molar ratio, the formation of a porous structure was reduced compared to the pure organic particles – potentially adding to the effect that the freeze-dried AS/SOM particles are not extremely efficient INPs.
  In our discussion of the freeze-drying experiment, we have only briefly mentioned the preactivation experiments by Wagner et al. (2017) on P16L497 (referred to the original version of the manuscript).

  We propose to extend this discussion as follows (lines 549ff):
  "Third, the porous organic material formed during the freeze-drying process could be a better ice nucleus on its own via the pore condensation and freezing mechanism **or by retaining an imprint of the sublimated ice on the highly viscous organic material.**  **Previous studies observed an efficient preactivation of glassy organic aerosol particles after ice-cloud processing (Wagner et al., 2012; Kilchhofer et al., 2021). Wagner et al. (2012) investigated in the AIDA chamber four different organic solutes, raffinose, 4-hydroxy-3-methoxy-DL-mandelic acid (HMMA), levoglucosan, and a multicomponent mixture of raffinose with five dicarboxylic acids and ammonium sulfate, and observed that the ice-cloud processed glassy aerosol particles catalyzed ice formation at ice saturation ratios between 1.05 and 1.30. Kilchhofer et al. (2021) confirmed the**

improved ice nucleation ability of cirrus-cloud processed raffinose particles in dedicated CFDC ice nucleation measurements. This raises the question of why the ice-cloud processed AS/SOM particles (observed ice onset at $S_{ice}$ = 1.41) do not reveal an even better ice nucleation ability.  However, repeated ice nucleation experiments with secondary organic aerosol particles produced from the ozonolysis of α-pinene performed by Wagner et al. (2017) showed a much lower susceptibility towards preactivation compared to the above mentioned organic solutes. For example, the ice nucleation onset of the ice-cloud processed α-pinene SOA particles at 220 K was at an ice saturation ratio of about 1.45. A similar ice nucleation threshold was found in an experiment where highly porous α-pinene SOA particles were formed by a freeze-drying process at 243 K (as in the current study) and then cooled to cirrus temperatures to probe their ice nucleation ability. Furthermore, Adler et al. (2013) showed that in mixed AS/OM particles with a 1:1 molar ratio, the formation of a porous structure was reduced compared to the pure organic particles, adding to the effect that the freeze-dried AS/SOM particles are not extremely efficient INPs."

- Regarding the phase of the AS fraction in the particles, the infrared spectrum of the freeze-dried a-pinene SOA particles in the regime of the $v_3$ ($SO_4^{2-}$) mode was similar to spectra A and C in Fig. 7c, indicating that the sulfate was effloresced.

  This is now mentioned in the revised manuscript, we added lines 526-528:
  "After the chamber cooling, the infrared spectrum of the freeze-dried AS/SOM particles in the regime of the $v_3$ ($SO_4^{2-}$) mode was similar to spectra A and C in Fig. 7c, indicating that the sulfate was effloresced."

**Minor/Technical Comments**
Title: I believe that "Ammonium, Sulfate, Secondary, and Organics" should all be lowercase.
We agree and corrected the title.

P2L38: To me, the phrase "allows to" here reads very awkward. Perhaps change to "allows us to"?
Done.

Equation 1: What was the prescribed density of organics in this work?
The density of the organic material is estimated to $1.25\ g/cm^3$. The density value is introduced at P7L209, referred to the original manuscript version (line 216 in the revised version).

P8L249: Here the phrase "allow to" shows up again and reads slightly awkward. I would suggest deleting it and changing "evacuate" to "evacuates".
Done.

P9L274: Perhaps change "thanks to" to "because of".
Thanks for the suggestion. Unfortunately, the sentence was wrong, now it reads (lines 287-288):
"Thus, a **because of the non-linear dependency of the equilibrium vapor pressure as a function of temperature, a** supersaturation profile establishes  **between the walls**."

P9L285: Perhaps change "allows to minimize" to "minimizes"

Done.

P10L305: Change "that" to "which".

Done.

P11L340: Perhaps change "reddish" to "red". I see that the symbols are slightly different in color, but, in this reviewer's opinion, it's okay to simple say "red".

P11L341: Perhaps change "greenish" to "green".

Yes, agreed and modified.

P13L415: The sentence "Note that in this specific experiment the chamber walls were not coated with ice" could be deleted. It feels extraneous to this reviewer.

The usual preparation procedure of the AIDA chamber described at P5L148 (in the original manuscript) includes the humidification step, that leads to the formation of a thin ice layer on the chamber walls. However, when ice is present in the chamber, it acts as a source of water vapor and does not allow to reduce the relative humidity, as needed for experiments of type D to induce the crystallization of the AS component. We think it is important to mention this experimental detail, but will extend our statement to better explain why we deviated here from the standard preparation procedure:

Lines 444-445 now read:

"Note that in this specific experiment the chamber walls were not  **ice-coated, because the ice layer would have acted as a source of water vapor and prevented the reduction of the relative humidity.**"

P14L434: Could cite the work of Freedman et al., 2020 here.

Thanks for the suggestion. The citation has been added.

P15L468: It seems as if the authors are suggested that the coated particles were not heterogeneously ice active, when Figure 5 seems to suggest they are. Perhaps the authors mean that they have higher onsets than homogeneous nucleation?

P16L518: Same comment as the one previous-the authors are implying that thickly coated particles are not "active as INPs".

Thank you for this comment. The ice nucleation onsets of the thickly SOM-coated AS particles are close to the homogeneous freezing threshold (Koop et al., 2000), but such particles are certainly not comparable to aqueous solution droplets for which the water-activity based parameterization for homogeneous freezing was developed. We will remove statements like that thickly coated particles are "not active as INPs" from the manuscript text and be more careful in our description.

Lines 511-513 become:

This might provide a pathway to  **increase the ice nucleation ability of** internally mixed AS/SOM particles with high organic mass fraction

particles, because a porous organic coating might less efficiently cover the ice  **nucleating** active sites of the crystalline AS component.

Lines 590-592 have been changed to:

"We performed two experiments to investigate whether there are pathways by which **the ice nucleation ability of** mixed AS/SOM particles with high organic weight fractions (25–50 wt%), as often found in the upper troposphere (Froyd et al., 2009), could  **be increased**."

**Corrigendum**

Additionally, we modified the manuscript in the following parts due to the presence of errors.

Lines 287-288:
Thus,  **because of the non-linear dependency of the equilibrium vapor pressure as a function of temperature, a** supersaturation profile establishes  **between the walls**.

Line 353:
"Effects" changed to "affects"

Line 429:
"Measurments" changed to "measurements"

We removed the sentence at lines 481-483, referred to the original version of the manuscript:

In fact, the change detected by the optical particle counter is probably related to the deliquesced ammonium sulfate component.

Figure 5b: in the legend the reference to "Ladino, 2016" has been changed to "Ladino, 2014".

References

**Adler, G.,** Koop, T., Haspel, C., Taraniuk, I., Moise, T., Koren, I., Heiblum, R. H. and Rudich, Y.: Formation of highly porous aerosol particles by atmospheric freeze-drying in ice clouds, Proc. Natl. Acad. Sci., 110(51), 20414–20419, doi:10.1073/pnas.1317209110, 2013.

**Kilchhofer, K.,** Mahrt, F. and Kanji, Z. A.: The Role of Cloud Processing for the Ice Nucleating Ability of Organic Aerosol and Coal Fly Ash Particles, J. Geophys. Res. Atmos., doi:10.1029/2020jd033338, 2021.

**Wagner, R.,** Möhler, O., Saathoff, H., Schnaiter, M., Skrotzki, J., Leisner, T., Wilson, T. W., Malkin, T. L. and Murray, B. J.: Ice cloud processing of ultra-viscous/glassy aerosol particles leads to enhanced ice nucleation ability, Atmos. Chem. Phys., 12(18), 8589–8610, doi:10.5194/acp-12-8589-2012, 2012.

**Wagner, R.,** Höhler, K., Huang, W., Kiselev, A., Möhler, O., Mohr, C., Pajunoja, A., Saathoff, H., Schiebel, T., Shen, X. and Virtanen, A.: Heterogeneous ice nucleation of {$\alpha$}-pinene SOA particles before and after ice cloud processing, J. Geophys. Res., 122(9), 4924–4943, doi:10.1002/2016JD026401, 2017.

---

## Author Comment (AC2)

We thank the anonymous Reviewer for the interesting and useful comments to improve the quality of our article.

Our replies are reported in the following with blue fonts, Reviewer's comments are reported in black italic fonts. Reporting the changes made in the manuscript, we indicate with strikethrough fonts the text previously present in the manuscript and now removed, and with bold fonts the additions we suggest to include in the revised manuscript. References to pages and lines refer to the revised version of the manuscript, unless noted otherwise.

**Major point 1:**

*The experiments conducted here in are done systematically with 5 different systems investigated (Experiments A – E). I think the discussion of cloud processing and its impacts have been kept too implicit. I think the authors could do a better job in addressing how cloud processing of organic coated AS can enhance the ice nucleation activity compared to unprocessed particles. The discussion is often presented from the other point of view, i.e. compared to pure crystalline AS, the cloud processed particles are less IN active, however, given the cloud cycling is a commonly occurring (REF) process (both through liquid and ice clouds) in the atmosphere, I recommend also making the comparison and bringing out the cloud processing aspect of the study more. Also, in the atmosphere more often than not, particles are internally mixed, as such their cloud processing studies would be most relevant for the troposphere.*

Thanks for this comment, we also believe that atmospheric aging and cloud cycles are important processes that need to be taken into consideration to correctly assess the ice nucleation ability of the particles. We now stress this aspect more strongly in several parts of the manuscript, and, as suggested, change the point of view and put more emphasis on the observation that the cloud-processed particles have an improved ice nucleation ability in comparison with the pure organic and organic-coated particles from experiments B and C.

In the Introduction, we further emphasize the importance of cloud processing at line 97:
"**Formation and dissipation of clouds, as well as cycles of humidification and drying are examples of processes commonly occurring in the atmosphere, but their effects on the physical properties of the so-processed aerosol particles are still poorly understood.**"

In the Introduction, lines 111-113:
Mixtures of organics and ammonium sulfate can also form at higher temperature and relative humidity conditions at which the particles are fully mixed**, as e.g. in liquid clouds**.

At the end of section 3.2 (Liquid cloud processing), we add lines 503-506:
"**The enhanced ice nucleation ability of the liquid cloud processed particles compared to the pure organic and organic-coated particles clearly indicates that the ice nucleation ability of atmospheric aerosol particles can strongly change during their lifetime in the atmosphere. Cloud processing of internally mixed aerosol particles is a common phenomenon, whose impacts on the particles' microphysical properties need to be investigated in future studies**."

At the end of Sect. 3.3 (Freeze-drying processing), we add lines 567-570:
"**In spite of the strong effect of thin SOM coating layers revealed by experiments of type C, we have shown that the ice nucleation ability of AS/SOM internally mixed particles is also**

**strongly linked to their thermodynamic history. Uplifting of particles to the upper troposphere via convective clouds, for example, could lead to particles with an enhanced ice nucleation ability.**"

In the Conclusions, we modified lines 613-616:
"**Since cloud processing is a common phenomenon, especially for particles uplifted in a convective system, their morphology changes and the enhancement of their ice nucleation ability need to be carefully characterized, also for internally mixed particles with different compositions (e.g., mineral dust particles coated with organic material).** "

**Major point 2:**
*Can the authors better justify their definitive claim that the particles experimented with had core-shell morphology in the paper, even in the case of the thick coatings in Experiment B. To me it seems like this is being taken for granted, but there is no definitive evidence being provided. Especially in Experiment B where the coatings are conducted at room temperature and low RH, glassy state for the SOM is favoured and thus how likely is it to form core shell morphology. While reading the paper, I think there is some evidence to possibly infer the presence of core-shell morphology, but the way the paper is written now, it is assumed from the get-go, that indeed this is the case, without leaving room doubt. I think at best the authors claim they believe it is core-shell morphology, or it should be clearly stated how they can definitively support this claim. I make suggestions below, of how such a claim could be inferred based on their ice nucleation results.*
*Line 311:* *the core-shell morphology at this point is stated with certainty. How can this be verified? Either provide more evidence that this is indeed the case, or the authors can suggest at this point that "we believe it to be core-shell morphology". See further below for how this can be inferred based on the arguments the authors already provide.*
Thank you for these remarks. We now more carefully indicate that we first only assume a core-shell morphology, and that the measured ice nucleation ability of the coated particles supports this hypothesis.

Abstract lines 13-15:
" **Small organic** mass fractions of 5-8 wt% **condensed on the surface of AS crystals** are sufficient to completely  **suppress the** ice nucleation ability of the inorganic  **component, suggesting that the organic coating is evenly distributed on the surface of the seed particles**."

Introduction lines 109-110:
"Particles investigated in experiments of type B and C **probably** had  **a** core-shell morphology, with the  **ice nucleation (IN) active** ammonium sulfate core shielded by the condensed  **not IN active** organic material."

Introduction lines 115-116:

"Would this process also inevitably lead to   **evenly coated particles where the heterogeneous IN activity of the AS component is strongly suppressed**?"

Sect. 3.1 Organic coating effect, lines 326-328:

"In both cases, **we assume that** the coating procedure led to particles with a core-shell morphology, with the solid AS as the core and the organic material surrounding it. **This hypothesis is supported by the ice nucleation measurements as discussed below."**

Sect. 3.1 Organic coating effect, we add lines 341-343:

**"The strong decrease of the particles' ice nucleation ability suggests that most of the IN active sites of the bare AS have been shielded by the condensed organic material distributed in a core-shell morphology."**

Sect. 3.1 Organic coating effect, lines 368-370:

"Experiments B1 and B2 have shown that thick SOM-coating layers almost completely suppress the heterogeneous ice nucleation ability of the AS  **component** and shift the particles' ice nucleation onset close to that observed for the pure SOA particles."

Sect. 3.1 Organic coating effect, we add lines 396-398:

**"The suppression of the particles' ice nucleation ability during and at the end of the coating procedure suggests, as for experiments of type B, that the organic material is evenly distributed on the AS surface and progressively covers its IN-active sites."**

Sect. 3.2 Liquid cloud processing, we modified lines 412-414:

"Experiments of type C, however, just tested one specific pathway for the formation of the mixed particles, where the crystalline AS was already present and then coated with the organic substances ."

Sect. 3.2 Liquid cloud processing, lines 417-418:

"Here, we started from thickly SOM-coated AS crystals  **as investigated in experiments of type B**, temporarily activated these particles to homogeneously mixed, aqueous droplets…"

Sect. 3.2 Liquid cloud processing, lines 462-463:

"The AIDA and INKA results indicate that the ice nucleation ability of the processed particles lies between those of the pure crystalline AS and of the thickly SOM-coated AS crystals ."

Sect. 3.2 Liquid cloud processing, we add lines 469-470:

**"This result again supports the hypothesis of a core-shell morphology for the coated particles of experiments B, for which the heterogeneous ice nucleation ability of the AS component was completely masked by the condensed organic material."**

Sect. 3.2 Liquid cloud processing, we add lines 476-479:
"The results for the pure SOA particles from Ladino et al. (2014) and for the thickly SOM-coated particles from this study agree, confirming that the coated particles have an ice nucleation behavior similar to the purely organic particles**, consistent with our assumption of a core-shell morphology**."

**Specific comments and suggestions:**

**_Line 49:_** _Change "accordance" to "agreement"_
Done.

**_Line 59:_** _Oxalic acid can also act as an INP in the deposition mode [Kanji et al., 2008]_
Thanks for this remark. We included the reference in the manuscript (lines 59-61):
"Some organic compounds, **e.g.**  oxalic acid, are able to crystallize and act as ice nucleating particles in the immersion freezing  (Zobrist et al., 2006; Wagner et al., 2015) **and deposition nucleation mode (Kanji et al., 2008)**."

**_Line 67:_** _could add Möhler et al. [2008] here as a reference for homogeneous freezing of pure SOM_
Done.

**_Line 68:_** _could add references Möhler et al. [2008] and Kanji et al. [2019] as examples of studies that used the same SOM as proxies for organic particles and coatings_
Done.

**_Line 106:_** _it maybe more accurate to say, "ice nucleation active" or "not ice nucleation active". Ice inactive is a little too colloquial and can have broader meaning._
**_Line 121:_** _consider using "ice nucleation active" instead of "ice-active"_
Thanks for this comment. We made the suggested change here and in other parts of the manuscript.

**_Line 108:_** _Delete comma after "humidity"_
Done.

**_Line 122-126:_** _here the authors describe Experiment D and refer to it as also forming liquid droplets, then why not just call this process liquid cloud processing. This would be more consistent with the terminology used in Experiment E. It is after all a process where liquid drops are formed first, followed by evaporation. So liquid cloud processing would be more suitable. Afterall the only difference between D and E is the absence of ice cloud formation. See comments on Figure 1._
**_Figure 1:_** _Why is experiment D not called liquid cloud processing, to be consistent with Experiment E which can be called cold cloud or ice cloud processing. The only difference between the two is freezing followed by sublimation vs. droplet activation followed by evaporation. Alternatively, D could be called liquid cloud processing and E could be called cloud processing_

*since E involves both liquid and ice. I would consider making the labelling more consistent. Even in the figure caption the authors state in the last line that E involves droplet activation thus it demonstrates the similarity between D and E (thus I would suggest both be part of a cloud processing label).*

Thanks for this suggestion. Experiments of type D are now called "liquid cloud processing". We adapted the text as follow:

In the schematic of Fig. 1 and in Table 1, the experiment of type D is now called "Liquid cloud processing".
The title of Section 3.2 was also changed: " **Liquid cloud processing** (experiment D)"

Line 416:
"This pathway was investigated with experiments of type D**, where liquid cloud processing is simulated**."

Lines 420-421:
Figure 7a shows a  **schematic** of the  **liquid cloud** process**ing** together with the possible phase states and morphologies of the particles during the experiment.

Lines 425-428:
Figure 7b presents the time series of the AIDA pressure (black line), AIDA temperature (red line), AIDA relative humidity with respect to water (blue line), and linear depolarization ratio measured with the SIMONE instrument (green line) during the  **liquid cloud processing** experiment.

Lines 603-604:
Our results from experiments D and E suggest that internally mixed particles, **that undergo**  **liquid or ice cloud** process**ing**, have unevenly distributed organic coating.

Caption of Figure 7:
 **Liquid cloud processing, re-crystallization,** and ice nucleation results (experiment type D).

**Line 176-177:** *Should read "The coating procedure in the APC chamber was performed at ambient temperature and low relative humidity and in the AIDA chamber at low temperature and ice saturated conditions."*
Done.

**Line 183:** Can the RH in the injection line be better quantified. For injection into AIDA 10% *RH*w at room temperature could be very high for the cold temperatures conducted below 233 K. Thus, this could imply that the particles deliquesced upon entering the AIDA chamber. Please clarify this point in the manuscript.

Thanks for this comment. 10% RH was indeed a very conservative estimate. To more accurately evaluate the humidity inside the injection line, we analyzed the increase in water vapor pressure inside the AIDA chamber during aerosol injection. We selected an experiment (not discussed in the article), where the AIDA chamber was controlled to $T = -50°C$ and $RH_w = 29\%$, i.e., ice-subsaturated conditions, because ice-coated walls would have acted as a sink for water vapor. The change in water vapor pressure measured by APicT during aerosol injection was $\Delta e = 1.857\ Pa - 1.837\ Pa = 0.02\ Pa$. The total volume of air injected through the sampling line was equal to $V_{inj} = 125\ L$. Thus, the measured change of water vapor pressure in the AIDA chamber ($V_{AIDA} = 84300\ L$) corresponds to a water vapor pressure in the injection line equal to $e = \Delta e\ \frac{V_{AIDA}}{V_{inj}} = 13.49\ Pa$, corresponding to a relative humidity lower than 1% in the injection line (at 25°C).

Additionally, the phase state of the particles was constantly monitored during aerosol injection with the FTIR measurements. In the FTIR spectra, we did not detect any signature of liquid water in the O-H stretching mode regime during and after aerosol injection (centered at 3300-3400 cm$^{-1}$), substantiating that the particles have not deliquesced.

We now use the estimate of $RH_w \leq 1\%$ in the manuscript (line 188).

***Line 197:*** *Delete the word "again"*
Line 204: Done.

***Line 222-223*** *where the coating thicknesses are given, can the RHw be given as well, to have an impression of the phase state of the SOM. The temperature is given in Table 1. Ideally the RHw could also be included in Table 1.*
Thanks for this suggestion. The RH$_w$ conditions at which the coating was performed are now included in Table 1.

***Line 237-238:*** *should read "In the backward direction, the polarization-resolved scatter light intensity is detected so that …"*
Line 250: Done.

***Line 256:*** *should read "The nucleated ice quickly grows to large crystals and…"*
Line 269: Done.

***Lines 281-285:*** *What is the temperature of the evaporation section set at. Is it set at the sample temperature, cold wall or warm wall temperature? This should be mentioned followed by the justification that the authors correctly address that the evaporation section will not influence the measurements in this study since it is conducted at cirrus conditions and any further ice nucleation in the evaporation section would require RHi to be larger than 100% which is not the case in the evaporation section. However, in the case liquid drops form in the growth section and if the evaporation section is at the cold wall temperature, then the drops could freeze before evaporating. This should not be of issue in the current study because of the low temperatures investigated, but should be clarified in terms of how the INKA chamber operates.*

Thanks for pointing this out, the evaporation section is set at the cold wall temperature. This information has been included in the revised version of the manuscript.

Lines 296-302 now reads:
"The lower third of the instrument is called droplet evaporation section. In this lower part of the instrument, sub-saturated conditions with respect to water are obtained by maintaining the walls at the same temperature **(equal to the cold wall temperature)**. The evaporation section  **minimizes** droplet and ice crystal coexistence when the instrument is operated at mixed-phase cloud conditions. When operated at cirrus conditions, the evaporation section does not influence the measurements because the environment is  at ice saturated conditions $(S_{ice} = 1)$, preventing the sublimation of the ice crystals. **Additionally, homogeneous freezing of solution droplets in the cold evaporation section is not expected to occur in the experiments presented here (T ≤ 45 °C)."**

***Line 314-315:*** *the thicknesses are calculated assuming uniform/even coating. This should be added to the end of the sentence. E.g. "…thickness **of** 44 nm assuming an even/uniform coating".*
Thanks for this remark, the text now reads (line 329-332):
"In experiment B1, we generated SOM-coated AS particles with an organic mass fraction of 26.9 % and a coating thickness  **of 28 nm**; in experiment B2, the organic mass fraction was 39.1 % and the coating thickness  **50 nm**. **As explained in Sect. 2.2, these coating thicknesses were computed for the median particle diameters assuming a uniform coating.**"

***Line 332:*** *replace "notified" with "noted"*
Line 351: Done.

***Line 335-339:*** *one may want to simply state that longer equilibration times are possible in the AIDA chamber compared to the CFDC, which is sort of stated, but not explicitly.*
Thanks for this suggestion. We now use the key-word equilibration time in the text.
Lines 355-356:
"During an AIDA expansion run, for example, the thermodynamic conditions inside the chamber change relatively slow  **and allow long equilibration times**."

***Lines 358-365:*** *What is shown in Figure 6a, that cannot be deduced from Figure 6b. The time perhaps, but this doesn't come up in any discussions or an anchor point for an argument. It looks like figure 6a is redundant and figure 6b nicely shows the point the authors are trying to make and with more clarity. I suggest removing Fig. 6a or make clearer what additionally it contributes to the discussion.*
***Figure 6:*** *As mentioned in the text above, the purpose of panel a is not clear to me. It could be removed. Also, in the text of the manuscript one could describe that experiment C nicely shows that gradually increasing the coating thickness progressively suppresses the IN activity. So far it is discussed as though this happens as a step function at 5%, but the experiments nicely show that this is a gradual process of Si increasing with organic mass fraction.*

We have removed Fig. 6a and included in Fig. 6b (Fig. 6a in the revised version of the manuscript) the activation curves for the thickly coated AS/SOM particles (experiments of type B).
We have removed the description of Fig. 6a (lines 356-361 in the original version of the manuscript) and adapted the text to properly describe Fig. 6.

We include lines 375-378:
"Figure 6a shows the **ice nucleating fraction (INF) as a function of the saturation ratio with respect to ice for humidity ramps performed at -54 °C with particles characterized by different organic mass fractions with INKA. The activation curves from experiments C1 and C3 are shown with solid symbols (circles and down-pointing triangles, respectively), data from experiments B1 and B2 are shown as empty circles.**"

And later we comment on the results, lines 383-386:
"**The ice nucleation ability of particles with ~8 wt% organic material (experiment C3) is almost identical to the one of the thickly coated particles from experiments B1 and B2 (with 27 wt% and 39 wt% organic mass fractions, respectively), indicating that such a low organic content, if uniformly distributed, is sufficient to suppress the ice nucleation ability of the solid AS particles.** "

Additionally, the gradual suppression of the AS ice nucleation ability as a function of the condensed organic material is more explicitly introduced in the text.

In the description of the results of Fig. 6b (previously 6c) we now state at lines 391-393:
"The data  confirm the results from  **the experiments presented in Fig. 6a**, showing that  **the ice nucleation ability of the solid AS component is gradually suppressed with the increase of organic material condensed on its surface**."

In the Conclusions, lines 579-582:
"However, we show that  **secondary organic material** condensed on the AS **surface progressively shifts the ice nucleation onset to higher ice saturation ratios. A small amount of SOM**, corresponding to an organic mass concentration of 5-8 wt%, is sufficient to  **increase** the ice nucleation onset of the coated particles to $S_{ice}$ > 1.45 (experiment of type C)."

**_Line 368-369:_** _The measurements definitely demonstrate the possibilities of combining CFDC with ageing experiments, but I don't think the statement about combining AIDA with CFDC is needed as mentioned in this sentence. AIDA is a complex chamber with highly sophisticated instruments coupled to it and capabilities. For the experiments being proposed here a chamber that can be cooled to low enough temperatures can be used here coupled with a CFDC. i.e. this is in reference to the step wise coating experiments in Experiment C. A chamber as complex and sophisticated as AIDA would not be necessary._
Thank you for this remark. We have removed that sentence.

***Line 370 – 377:*** *Could the authors also comment on the onset Si in Fig 6c for experiment C1-C3 at the same organic coating % i.e. why is the onset Si lower for the warmer T experiments than the colder T experiments at the same organic coating %? From ice nucleation perspective the ice germ should be smaller at colder T, so the onset for colder T should be at lower Si or the same. Is it a diffusional growth limitation to when the OPC detects ice crystals or is it a limitation of diffusing water vapour into the organic coating? Some comment addressing the difference in onset Si for the same organic coating percent as a function of temperature should be made here.*

Thanks for this interesting comment. Temperature-dependent ice nucleation onsets for α-pinene SOA have also been measured e.g. by Wagner et al. (2017) and Charnawskas et al. (2017). Ice nucleation onsets for α-pinene SOA from Wagner et al. (2017) are also shown in Fig. 5b in the manuscript. These ice nucleation onsets and their temperature trend are very similar to those determined for aqueous sulfuric acid solution droplets, whose freezing behavior is described by the water activity-based homogeneous freezing parameterization (Koop et al., 2000). This could indicate that at least the outer layer of the organic material has liquefied during the particles' residence time in INKA, making homogeneous freezing a suitable ice nucleation pathway, and explaining the temperature trend (increase of $S_{ice}$ with decrease in temperature). We also need to take into consideration that the condensed organic material could have a different chemical composition and viscosity at the different temperatures of the in situ coating experiments (Huang et al., 2018), thereby affecting the water diffusion into the particles and the ice nucleation pathways (Berkemeier et al., 2014; Lienhard et al., 2015; Price et al., 2015; Fowler et al., 2020).

We have included this discussion point also in the manuscript, lines 399-406:
"**The temperature trend of the ice nucleation onsets for organic mass fractions larger than about 4 wt% revealed by experiments C, i.e., higher $S_{ice}$ values with decreasing temperature, could point to a homogeneous ice nucleation pathway of the coated particles, meaning that at least the outer layer of the organic material has liquefied during the particles' residence time in INKA. A similar temperature trend has been observed in ice nucleation studies with pure α-pinene SOA particles (e.g, Ladino et al., 2014; Wagner et al., 2017; Charnawskas et al., 2017). Furthermore, we need to consider that the condensed organic material could have a different chemical composition and viscosity at the different temperatures of the in situ coating experiments (Huang et al., 2018), thereby affecting the water diffusion into the particles and the ice nucleation pathways (Berkemeier et al., 2014; Lienhard et al., 2015; Price et al., 2015; Fowler et al., 2020).**"

***Line 421:*** *should read "….with INKA at -54 °C immediately…"*
***Line 422:*** *delete "at -54 °C"*
Lines 451-452: Done.

***Line 424:*** *having the diffusion dryer upstream of the INKA instrument, could have modified the phase state of the organic from liquid to glassy, as such would it not be correct to state that INKA is already sampling glassy or phase separated aerosol since the LLPS occurs at RHw < 90%.*
The presented INKA measurements were performed after the crystallization of the AS component, when the AIDA conditions were $T = -5°C$ and $RH_w \approx 32\%$, conditions at which the SOM is expected to be already glassy (Petters et al., 2019). Of course, the dryer has further

decreased the relative humidity, but if the particles have already been glassy at $RH_w \approx 32\%$, the further reduction in $RH_w$ should not have altered the particle phase state. For experiment D2, we also performed an INKA scan after the overnight cooling of the AIDA chamber to $-50°C$, thus without the need of a diffusion dryer. The measured activation curve was very similar to the one obtained immediately after crystallization with the use of a dryer.

We modified the text as follows (lines 453-458):
"As the INKA measurements were performed by sampling from the AIDA chamber at $-5°C$ **and** $RH_w \approx 32\%$, a diffusion dryer was used to prevent frost formation in the instrument inlet. **To infer the possible effect of the dryer on the ice nucleation ability of the recrystallized particles, an additional INKA measurement was performed after the cooling of the AIDA chamber, thus without the need of a diffusion dryer. The ice nucleation ability measured with and without the dryer (i.e., before and after the cooling) are comparable (not shown), indicating that the dryer did not influence the ice nucleation ability of the particles.**

**_Line 423-434:_** _This argument presented here is quite interesting and can be exploited to support the authors' claim that core-shell morphology or complete coating was achieved in the thick coating experiments (B). The reason being that Experiment B completely suppressed the het. ice nucleation signal compared to that of pure AS in experiment A. Then coatings as thick as those in Experiment B but in Experiment D (liquid cloud processing) resulted in intermediate IN activity. As the authors point out that this must have resulted in changes in morphology allowing the initially thickly coated particles to become ice nucleation active after the liquid processing. This observation together with Experiment B and A would then more concretely support the core-shell morphology of the thick coatings._
_I think before this point, it is hard to lay claim to it so definitively without more concrete evidence._
Thanks for this comment. We have included this observation in the revised version of the manuscript. The intermediate ice nucleation ability of experiments of type D, indeed, support the hypothesis of fully covered AS seeds in the thickly-coated experiments (type B).
As already mentioned in the response to "Major comment 2", we have made the following changes:

Sect. 3.2 Liquid cloud processing, we add lines 468-469:
**"This result again supports the hypothesis of a core-shell morphology for the coated particles of experiments B, for which the heterogeneous ice nucleation ability of the AS component was completely masked by the condensed organic material."**

Sect. 3.2 Liquid cloud processing, we add lines 476-479:
"The results for the pure SOA particles from Ladino et al. (2014) and for the thickly SOM-coated particles from this study agree, confirming that the coated particles have an ice nucleation behavior similar to the purely organic particles**, consistent with our assumption of a core-shell morphology**."

**_Line 451:_** _I think the authors should specify that they mean > 5% by coating mass when they say "even thin coatings" because below 5% there was ice nucleation activity below the_

*homogeneous freezing threshold suggesting that coatings with mass % < 5% will not completely suppress the het. IN activity.*

Thanks for this comment, we agree with the Reviewer and changed the manuscript as follows (lines 487-489):
"The finding that  α-pinene  **SOM mass fractions greater than ~5-10 wt% completely suppress** the heterogeneous ice nucleation ability of crystalline AS (Sect. 3.1) rules out that immersion freezing is a prevalent nucleation mode in our experiments."

***Line 456:*** *why would the crystalline AS core not dissolve. It was not clear to me how immersion freezing by the crystalline core is being explained here. Shouldn't the AS core dissolve in water if it is immersed?*

The case suggested here refers to AS immersed in a liquid organic component, where the AS will remain solid until its deliquescence RH is exceeded. The aqueous organic layer, if not glassy/highly viscous, allows the solid core to act as ice nucleating entity in the immersion freezing mode (Schill et al., 2014; Schill and Tolbert, 2013). Schill et al. (2014) nicely illustrate the deliquescence and ice nucleation behavior of aqueous SOM/AS particles in their Figs. 5 and 6. For temperatures above 230 K, the deliquescence of AS, not present as consolidated crystals but distributed in the form of many "islands", is observed at $RH_w$ of about 80%. At temperatures below 230 K, heterogeneous ice nucleation was observed before surpassing the deliquescence relative humidity, ascribed to immersion freezing of the aqueous organic phase induced by the AS islands. Immersion freezing under cirrus conditions is different to immersion freezing under mixed-phase cloud conditions because it occurs below liquid water saturation.

We have now included the references to Schill et al. (2014) and Schill and Tolbert (2013) in the text (lines 486-487):
"This activity may be controlled by the viscosity and water solubility of the organic material **(Schill and Tolbert, 2013; Schill et al., 2014)."**

***Line 461-462:*** *Why would AS efflorescence be inhibited at low temperatures, can this be explained more clearly to make the connection to the point above this sentence.*

Bodsworth et al. (2010) suggested that ammonium sulfate particles rich in organics do not reach a large supersaturation with respect to ammonium sulfate at low temperatures, possibly due to non-ideal solution behavior, and thus inhibiting the homogeneous nucleation of the AS crystal. The authors also proposed a possible connection to the viscosity of the organic material, citric acid in their study. For example, an organic mass fraction of 0.33 was sufficient to inhibit ammonium sulfate efflorescence at temperatures below 250 K (Figs. 4 and 5 in Bodsworth et al. (2010)). If internally mixed AS/SOM particles experience a low RH at low temperatures (in contrast to the −5℃ condition presented in this study) and the efflorescence of the AS component is inhibited, this would preclude the formation of partially engulfed structures with a not fully coated AS component, meaning that liquid-cloud processing would not increase the ice nucleation ability of the particles.

The revised manuscript now reads (lines 498-502):
"In particular, at lower temperatures and/or higher amounts of organics, the AS efflorescence could also be inhibited (Bodsworth et al. (2010)). **If re-crystallization of the AS component and the formation of a partially engulfed particle morphology did not occur at lower temperatures (in contrast to the -5°C condition simulated in this study), liquid cloud processing would not increase the ice nucleation ability of the AS/SOM particles. This behavior could be investigated in future AIDA experiments where the re-crystallization process is performed at lower temperatures.**"

**_Line 479:_** _The authors should be careful in describing the freeze-drying process. After activation into drops and freezing, just completes the freezing part of the freeze drying, and then upon sublimation comes the drying part, as such saying that all aerosol particles underwent freeze-drying in line 479 is still too early. It is only after line 481 where the authors state "...thereby quickly sublimated" is when the freeze-drying process has been accomplished._
Thanks for this remark, we have corrected the sentence (lines 522-523):
"This additional fast drop in pressure almost instantly reduced the gas temperature by another 2.5 K and caused the entire droplet population to freeze homogeneously, ."

**_Line 499._** _Here one more discussion point could be added is the how cloud processing enhances ice nucleation of organic coated particles. This is not brought out explicitly enough. See major comment 1._
Thanks for this comment. As already reported in the reply to major comment 1, we have now added a paragraph at the end of Sect. 3.2 where we explicitly mention the enhancement of the IN ability of the processed particles.

Lines 503-506:
"**The enhanced ice nucleation ability of the liquid cloud processed particles compared to the pure organic and organic-coated particles clearly indicates that the ice nucleation ability of atmospheric aerosol particles can strongly change during their lifetime in the atmosphere. Cloud processing of internally mixed aerosol particles is a common phenomenon, whose impacts on the particles' microphysical properties need to be investigated in future studies**."

**_Line 505_** _and elsewhere where troposphere is mentioned. A mention that the SOM here may not be representative of tropospheric SOM is warranted because almost always photochemistry would play a role in the nature of SOM in the atmosphere, and here no UV source is mentioned in the production of SOM. This should be acknowledged somewhere in the manuscript. I suggest once in the methods where the SOM production is described and then again in the conclusion section._
Thanks for this interesting remark, Piedehierro et al. (2021) have recently investigated the ice nucleation ability of α-pinene SOA particles at temperatures below 243 K as a function of the formation mechanism and the RH conditioning. Particles were formed in a potential aerosol mass reactor either via dark ozonolysis or by forming OH radicals first. Their results show that the two different reaction pathways do not influence the ice nucleation ability of the formed particles. The freezing onsets of the SOA particles were detected at or above the homogeneous

freezing threshold for aqueous solution droplets, regardless of the formation mechanism. This suggests that the experiments presented in our study are not influenced by the α-pinene oxidation mechanism.

As suggested by the Reviewer, the possible effect of the nucleation and growth mechanism for SOM derived from α-pinene is now mentioned in Sect. 2.2 (Aerosol preparation and characterization) and in the conclusions.

Lines 190-192:
"**Another important pathway for the formation of α-pinene SOM in the atmosphere are photo-oxidation reactions, but the ice nucleating ability of the organic material generated via dark ozonolysis or photo-oxidation of α-pinene was found to be very similar (Piedehierro et al., 2021)**."

Lines 585-587:
"**The ozonolysis of α-pinene is not the only mechanism responsible for upper tropospheric SOM, as photolysis also plays a crucial role during daytime. However, the nucleation and growth mechanisms does not influence the ice nucleation ability of α-pinene derived organic material (Piedehierro et al., 2021)**."

___**Line 510-513:**___ *this part ignores the fact that coating mass % below 5% still yield het. Ice nucleation active particles. This should be acknowledged here.*
We now more clearly state that the condensed organic material gradually suppresses the IN ability of the bare AS.

Lines 579-582:
"However, we show that  **secondary organic material** condensed on the AS **surface progressively shifts the ice nucleation onset to higher ice saturation ratios. A small amount of SOM**, corresponding to an organic mass concentration of 5-8 wt%, is sufficient to increase the ice nucleation onset of the coated particles to $S_{ice}$ > 1.45 (experiment of type C)."

___**Line 527:**___ *should read "(i.e., ice nucleation ability between pure AS and AS with a compact…)"*
Done (line 602).

**Figures and Tables**

___**Figure 1:**___
*In the figure caption "upper tropospheric conditions" is mentioned, but I think to be more specific the authors should say upper tropospheric temperature, since photolysis is not part of this work, or UV is not been simulated in this study which would be part of UT conditions.*
Thanks for this note. We have changed the caption of Fig. 1 as suggested.

___**Table 1:**___ *O3 concentration can be changed to ppm and scientific notation removed and all numbers harmonised in terms of how many significant digits are given. Significant digits can also be harmonised for all experiments of α-pinene concentrations.*

Done.

*Figure 3.* *Would be helpful for reading if more than 3 ticks are added to the x-axis of panels a and b. i.e. every 0.1 μm and also add the scale on the x-axis of panel a.*
Done.

*Figure 5: panel a, either label all experiments as Exp A/B1/B2 or then AS, AS/SOM thick coating etc. but currently it is switched from AS then EXP B1/B2. Choose one style to be consistent.*
*Panel b. the contrast between the literature results in red and the orange data from the current study isn't that great. Also, the distinction between the green from the current study and literature is not that clear either. To make this clearer, I would suggest keeping the data from this study in black since AIDA and INKA are already shape differentiated. This will make it very clear to a viewer of where and how the data from this study compare to the literature.*
Thanks for these suggestions, Fig. 5 was changed as proposed.

*Figure 7. One could consider another panel, where the activation curves for Exp A1, and 3 mass fractions of C2 and B1 are plotted, to show the progressive nature and that experiments with coatings larger than 5% yield the same result as that of ~30%.*
This plot has already a lot of information and the focus is on the processed particles (experiments of type D). However, it is a good suggestion to compare the thin coating experiments to the thick coating ones. We added the activation curves for experiments B1 and B2 to Fig. 6a.

*Figure 8. what are the different blue coloured dots in panel b? This is not described in the caption.*
The blue dots are the single particle data from the welas optical particle counters (OPC1 and OPC2), each dot corresponds to an aerosol particle, cloud droplet or ice crystal at the corresponding size. The caption has been updated:
"Lower panel:  **single** particle  data **from the welas optical particle counters (OPC1 and OPC 2),** each dot corresponds to an aerosol particle, droplet, or ice crystal at the corresponding size."

*Also, the green curve crossing the yellow curve in panel D is not discussed or addressed anywhere in the text. This crossing suggests that at a certain saturation the thickly coated particles are more active than the cloud processed particles (even if it is in the homogeneous freezing regime).*
This is a nice consideration, thank for pointing this out. Although this observation will need further future evaluation, we have included in the text a possible interpretation.

From line 536:
"**The activation curve of the freeze-dried particles measured with INKA (Fig. 8d, yellow data) has a different profile compared, for example, to the thickly-coated particles (green data). The slower increase of the ice nucleating fraction as a function of the ice saturation ratio (similar to the activation curve for AS in orange) suggests that heterogeneous freezing is the dominant ice nucleation mechanism occurring in INKA up to $S_{ice} \sim 1.6$. This effect could be related to**

the limited residence time of the particles in the INKA instrument, which together with a slower water uptake from the particles could have shifted the detected homogeneous freezing onset to higher $S_{ice}$ values. This kinetic limitation is not evident in the AIDA data, probably due to longer equilibration time during the expansion run. Particles in experiment E1 will likely be more viscous than particles from experiment B2 due to their larger organic mass fraction and/or due to a change of the chemical-physical properties of the particles after the freeze-drying process."

**Corrigendum**
Additionally, we modified the manuscript in the following parts due to the presence of errors.

Lines 287-288:
Thus,  **because of the non-linear dependency of the equilibrium vapor pressure as a function of temperature, a** supersaturation profile establishes  **between the walls**.

Line 353:
"Effects" changed to "affects"

Line 429:
"Measurments" changed to "measurements"

We removed the sentence at lines 481-483, referred to the original version of the manuscript:

In fact, the change detected by the optical particle counter is probably related to the deliquesced ammonium sulfate component.

Figure 5b: in the legend the reference to "Ladino, 2016" has been changed to "Ladino, 2014".

References:

Berkemeier, T., Shiraiwa, M., Pöschl, U. and Koop, T.: Competition between water uptake and ice nucleation by glassy organic aerosol particles, Atmos. Chem. Phys., 14(22), 12513–12531, doi:10.5194/acp-14-12513-2014, 2014.

Bodsworth, A., Zobrist, B. and Bertram, A. K.: Inhibition of efflorescence in mixed organic-inorganic particles at temperatures less than 250 K., Phys. Chem. Chem. Phys., 12(38), 12259–12266, doi:10.1039/c0cp00572j, 2010.

Charnawskas, J. C., Alpert, P. A., Lambe, A. T., Berkemeier, T., O'Brien, R. E., Massoli, P., Onasch, T. B., Shiraiwa, M., Moffet, R. C., Gilles, M. K., Davidovits, P., Worsnop, D. R. and Knopf, D. A.: Condensed-phase biogenic-anthropogenic interactions with implications for cold cloud formation, Faraday Discuss., 200, 165–194, doi:10.1039/c7fd00010c, 2017.

Fowler, K., Connolly, P. and Topping, D.: Modelling the effect of condensed-phase diffusion on the homogeneous nucleation of ice in ultra-viscous particles, Atmos. Chem. Phys., 20(2), 683–698, doi:10.5194/acp-20-683-2020, 2020.

Huang, W., Saathoff, H., Pajunoja, A., Shen, X., Naumann, K. H., Wagner, R., Virtanen, A., Leisner, T. and Mohr, C.: α-Pinene secondary organic aerosol at low temperature: Chemical composition and implications for particle viscosity, Atmos. Chem. Phys., 18(4), 2883–2898, doi:10.5194/acp-18-2883-2018, 2018.

Koop, T., Luo, B., Tsias, A. and Peter, T.: Water activity as the determinant for homogeneous ice nucleation in aqueous solutions, Nature, 406(6796), 611–614, doi:10.1038/35020537, 2000.

Ladino, L. A., Zhou, S., Yakobi-Hancock, J. D., Aljawhary, D. and Abbatt, J. P. D.: Factors controlling the ice nucleating abilities of α -pinene SOA particles, J. Geophys. Res. Atmos., 119(14), 9041–9051, doi:https://doi.org/10.1002/2014JD021578, 2014.

Lienhard, D. M., Huisman, A. J., Krieger, U. K., Rudich, Y., Marcolli, C., Luo, B. P., Bones, D. L. and Reid, J. P.: Viscous organic aerosol particles in the upper troposphere : diffusivity-controlled water uptake and ice nucleation ?, , 13599–13613, doi:10.5194/acp-15-13599-2015, 2015.

Petters, S. S., Kreidenweis, S. M., Grieshop, A. P., Ziemann, P. J. and Petters, M. D.: Temperature- and Humidity-Dependent Phase States of Secondary Organic Aerosols, Geophys. Res. Lett., 46(2), 1005–1013, doi:10.1029/2018GL080563, 2019.

Piedehierro, A., Welti, A., Buchholtz, A., Korhonen, K., Pullinen, I., Summanen, I., Virtanen, A. and Laaksonen, A.: Ice nucleation on surrogates of boreal forest SOA particles: effect of water content and oxidative age, Atmos. Chem. Phys., 1–17, doi:10.5194/acp-2021-10, 2021.

Price, H. C., Mattsson, J., Zhang, Y., Bertram, A. K., Davies, J. F., Grayson, J. W., Martin, S. T., O'Sullivan, D., Reid, J. P., Rickards, A. M. J. and Murray, B. J.: Water diffusion in atmospherically relevant α-pinene secondary organic material, Chem. Sci., 6(8), 4876–4883, doi:10.1039/C5SC00685F, 2015.

Schill, G. P. and Tolbert, M. A.: Heterogeneous ice nucleation on phase-separated organic-sulfate particles: Effect of liquid vs. glassy coatings, Atmos. Chem. Phys., 13(9), 4681–4695, doi:10.5194/acp-13-4681-2013, 2013.

Schill, G. P., De Haan, D. O. and Tolbert, M. A.: Heterogeneous ice nucleation on simulated secondary organic aerosol, Environ. Sci. Technol., 48(3), 1675–1682, doi:10.1021/es4046428, 2014.

Wagner, R., Höhler, K., Huang, W., Kiselev, A., Möhler, O., Mohr, C., Pajunoja, A., Saathoff, H., Schiebel, T., Shen, X. and Virtanen, A.: Heterogeneous ice nucleation of {$\alpha$}-pinene SOA particles before and after ice cloud processing, J. Geophys. Res., 122(9), 4924–4943, doi:10.1002/2016JD026401, 2017.